# Planning with Quantized Opponent Models

**Xiaopeng Yu**    **Kefan Su**    **Zongqing Lu**[*]

School of Computer Science, Peking University

## Abstract

Planning under opponent uncertainty is a fundamental challenge in multi-agent environments, where an agent must act while inferring the hidden policies of its opponents. Existing type-based methods rely on manually defined behavior classes and struggle to scale, while model-free approaches are sample-inefficient and lack a principled way to incorporate uncertainty into planning. We propose Quantized Opponent Models (QOM), which learn a compact catalog of opponent types via a quantized autoencoder and maintain a Bayesian belief over these types online. This posterior supports both a belief-weighted meta-policy and a Monte-Carlo planning algorithm that directly integrates uncertainty, enabling real-time belief updates and focused exploration. Experiments show that QOM achieves superior performance with lower search cost, offering a tractable and effective solution for belief-aware planning.

## 1    Introduction

Opponent modeling is considered an important aspect of intelligent behavior in partially observable multi-agent environments[1]. Early robotics and game-AI systems constructed explicit *type libraries*—small sets of handcrafted behavior classes—and then fitted the observed action stream to the most plausible type online [36, 47]. Advances in machine learning broadened this agenda, giving rise to statistical recognizers that detect plays [44], formations [13], or higher-level strategies in real-time strategy games [39, 49]. Complementary model-free approaches forgo explicit opponent representations and instead adapt the agent's policy through reinforcement learning [2]; yet such adaptation is sample-inefficient and brittle when observations are noisy or delayed. Crucially, regardless of whether the agent maintains explicit opponent beliefs, it must still *plan ahead* under uncertainty. Monte-Carlo Tree Search (MCTS) [8] is the de-facto planner for large sequential domains, but its branching factor explodes when every node must marginalize over many hypothetical opponent moves. Naïve belief propagation inside the tree quickly becomes intractable [9, 14]. Bridging expressive opponent modeling with tractable belief-aware planning therefore remains an open challenge.

We address this challenge by proposing *Quantized Opponent Models* (QOM). Our core insight is to compress the vast policy space of potential opponents into a finite, interpretable catalog of latent *types* discovered offline by a quantized autoencoder. During online interaction, the agent maintains a Bayesian belief over these types; by using a pre-trained decoder, the model can compute likelihoods efficiently enough to support real-time updates. This belief is used to sample opponent types in simulation, mix agent policies into a soft best-response *meta-policy*, and guide planning through a belief-informed prior. Consequently, every rollout (i) samples opponent actions from the current posterior, (ii) selects agent actions that are promising under that posterior, and (iii) refines the belief along the simulated trajectory, allowing planning and inference to reinforce each other.

Our framework thus unifies three previously disparate strands of research. From type-based opponent modeling, it inherits interpretability while discarding manual type design; from Bayesian reasoning

---

[*]Correspondence to ✉ zongqing.lu@pku.edu.cn

39th Conference on Neural Information Processing Systems (NeurIPS 2025).

it inherits principled uncertainty handling without incurring prohibitive inference costs; and from Monte-Carlo planning it inherits strong look-ahead search while injecting belief information directly into the tree policy. We provide theoretical analysis of posterior concentration under uncertain opponents, and empirical results demonstrating competitive performance gains over state-of-the-art baselines in benchmark games with partial observability and adversarial dynamics.

## 2 Related Work

Type-based opponent modeling has evolved from early behavior classification schemes that used handcrafted feature predicates for adversarial domains [36, 47]. Subsequent work explored declarative and similarity-based formulations to transfer knowledge across game instances [48, 39], while approaches extracted relational or sequential patterns to anticipate high-level strategies in both real-time strategy games and sports domains [54, 6, 31]. Researchers further demonstrated online adaptation of offensive tactics [27, 26] and recognition of group activities from multimodal trajectories [44, 28, 38]. More recent studies integrate positional data streams to identify formations on-the-fly or learn objective-driven agent behaviors from logged encounters [13, 20]. Recent hierarchical formulations have also been proposed to couple opponent modeling with planning in structured goal spaces [16]. Collectively, these works substantiate the effectiveness of representing opponents by a finite catalog of "types"—an idea our Quantized Opponent Models embrace, while unifying type discovery and belief maintenance within a single quantized autoencoder framework.

Bayesian reasoning provides a complementary perspective, treating the opponent's policy (or latent parameters) as a random variable and continuously updating posterior beliefs as evidence accrues. Early vision-based systems inferred multi-agent activities through dynamic Bayesian networks [17, 18], and robot-soccer agents leveraged probabilistic motion models to predict trajectories and openings [37, 4]. Subsequent work applied hierarchical or domain-specific priors to forecast build orders in real-time strategy games [49, 51] and to locate sweet spots in racket sports [55]. Recent advances scale Bayesian inference to negotiation [7], normative coordination [34], inverse games [30], and safety-aware exploitation under uncertain type beliefs [29]. Our framework inherits this probabilistic tradition by maintaining an explicit posterior over latent policy types and amortizes likelihood computation to enable real-time updates during planning rollouts.

MCP, especially MCTS, has become a widely adopted method for look-ahead decision making. Foundational variants such as Bayes-adaptive and information-set MCTS [9, 14] inspired extensions to multi-agent POMDPs [3, 21] and typed or open systems [35, 12]. Recent research refines value backup and exploration principles [11], reuses beliefs across rollouts [50], adapts models online [60], and incorporates learning for path-planning or decentralized coordination [46, 10, 59]. Risk-aware shielding, decision-theoretic communication, and nested MCTS methods further widen the algorithmic toolbox [32, 19, 43]. Our method builds on this line of work by incorporating QOM's belief-weighted meta-policy and Bayesian belief updates directly into the agent action selection, thereby synthesizing advances in type abstraction, probabilistic reasoning, and MCP into a unified belief-aware planning algorithm.

## 3 Method

In this section, we introduce QOM for planning in multi-agent partially observable environments. Our key idea is to combine a compact, discrete representation of opponent policies with principled belief tracking and planning under the unknown opponent policy. We first describe the QOM that captures opponent behavior, and then outline how to construct a meta-policy based on the learned latent types, and finally detail the integration of this belief model into an MCP framework for planning. The algorithm architecture is summarized in Figure 1.

### 3.1 Preliminaries

We consider a partially observable stochastic game [15], defined as a tuple $\mathcal{G} = (\mathcal{S}, \mathcal{A}_i, \mathcal{A}_{-i}, \mathcal{O}_i, \mathcal{O}_{-i}, \mathcal{P}, O, R_i, \gamma)$, involving an agent $i$ and a set of other agents (opponents) $\mathcal{N}_{-i}$. We treat all opponents jointly as a single composite agent $-i$ with joint action space $\mathcal{A}_{-i}$ and joint observation space $\mathcal{O}_{-i}$, where $\mathcal{S}$ is the hidden state space, $\mathcal{A}_i$ and $\mathcal{A}_{-i}$ are the action spaces of agent $i$ and the joint opponent, $\mathcal{O}_i$ and $\mathcal{O}_{-i}$ are the observation spaces, $\mathcal{P}(s' \mid s, a_i, a_{-i})$ is the transition

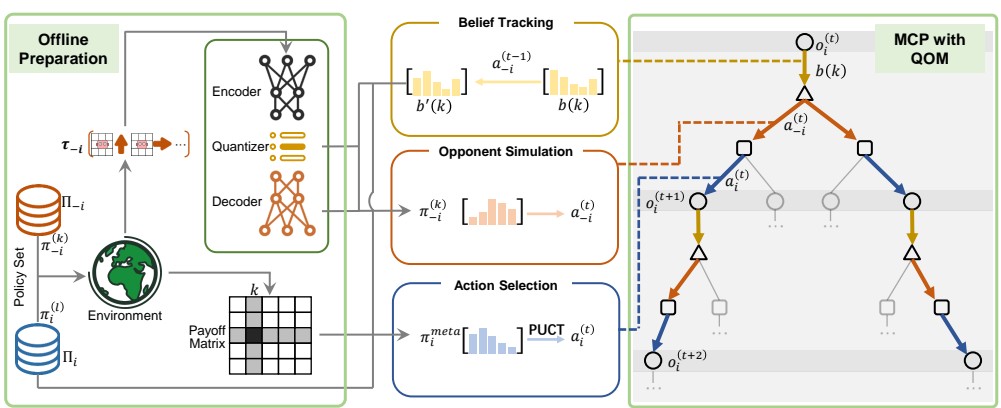

Figure 1: Overview of the QOM framework. The agent maintains a belief over latent opponent types, which is updated based on observed actions. The quantized autoencoder learns a discrete representation of opponent policies, and the meta-policy is constructed as a mixture of soft best responses to these types. This meta-policy is then used for planning.

probability, $O_i(o_i \mid s')$ and $O_{-i}(o_{-i} \mid s')$ are observation models, $R_i(s, a_i, a_{-i})$ is the reward for agent $i$, $\gamma \in [0, 1)$ is the discount factor.

At each time step $t$, both agent $i$ and the opponent select actions $a_i^{(t)}$, $a_{-i}^{(t)}$ based on their own local observation-action histories $h_i^{(t)}$, $h_{-i}^{(t)}$. The agent's goal is to maximize the expected return:

$$J_i = \mathbb{E}_{\pi_i, \pi_{-i}} \left[ \sum_{t=0}^{\infty} \gamma^t R_i \big(s^{(t)}, a_i^{(t)}, a_{-i}^{(t)}\big) \right]. \tag{1}$$

Besides the game definition, following resources are required: (i) an *agent policy library* $\Pi_i = \{\pi_i^{(l)}\}_{l=1}^L$ that stores a set of candidate policies; (ii) an *opponent policy library* $\Pi_{-i} = \{\pi_{-i}^{(j)}\}_{j=1}^J$ that supplies diverse behaviors; and (iii) an *environment simulator* that produces transitions $(s', o, r) \sim \mathcal{G}(s, a_i, a_{-i})$ and is used both for offline data generation and for online rollouts during planning.

## 3.2 Quantized Opponent Models

To reason effectively in the presence of an unknown opponent, we introduce the Quantized Opponent Models (QOM), which models the opponent's policy using a discrete latent variable representation. QOM approximates the opponent's potentially complex policy using a set of representative policy "types," learned via a quantized autoencoder (VQ-VAE) [52] where the raw policy sequences are modeled autoregressively.

**Latent opponent types via quantized autoencoder.** We approximate the space of opponent policies with a finite set of latent *types* $\{1, \ldots, K\}$. An offline-trained quantized autoencoder maps each opponent trajectory $\tau$, collected by letting an agent policy $\pi_i^{(l)}$ interact with an opponent policy $\pi_{-i}^{(j)}$, to a continuous embedding $z = f_{\text{enc}}(\tau) \in \mathbb{R}^d$, then assigns it to the nearest codebook vector $e_k$. The decoder $f_{\text{dec}}(e_k, \cdot)$ produces a step-wise likelihood $\tilde{\pi}_k(a_{-i} \mid h)$, enabling both fast simulation and likelihood evaluation. This discretization turns an intractable policy space into a tractable, interpretable catalog of opponent policies.

**Belief over latent types.** The belief $b_t(k)$ is defined as the probability that the opponent follows type $k$; it forms a categorical distribution over the $K$ latent types. All opponent-dependent quantities, opponent quantized policy $\tilde{\pi}_k$ and rollout priors, are averaged under this distribution, so planning automatically balances the risks of different hypotheses. A broad belief encourages exploratory behavior, whereas a sharply peaked belief lets the agent exploit its current inference. Thus $b_t(k)$ is the single latent variable that ties together opponent modeling and decision making.

### 3.3 Meta-Policy Construction

Given the belief distribution $b_t(k)$ over latent opponent types and a set of agent policies $\Pi_i$, we define a meta-policy for the agent to use during rollouts. The meta-policy is constructed as a belief-weighted mixture of soft best responses, reflecting the agent's expected performance against different opponent types. For each opponent type $k$, we compute a soft response distribution over the agent policy set using a softmax rule:

$$\sigma(\pi_i^{(l)} \mid k) = \frac{\exp(R_{l,k})}{\sum_{l'} \exp(R_{l',k})}, \tag{2}$$

where $\beta$ is temperature parameter and $R_{l,k}$ is the estimated expected return of agent policy $\pi_i^{(l)}$ when interacting with opponent type $k$. These values can be approximated via offline simulations, and represent a kind of *payoff matrix* between the agent and latent opponent policies.

Intuitively, $\sigma(\pi_i^{(l)} \mid k)$ represents how suitable agent policy $\pi_i^{(l)}$ is as a response to a particular opponent type $k$. A higher $R_{l,k}$ results in a higher selection probability, which controls how sharply the distribution concentrates around the best-responding policies.

The agent's overall rollout policy is then a belief-weighted mixture over all opponent types and corresponding soft responses:

$$\pi_i^{\text{meta}}(a_i \mid h_i) = \sum_{k=1}^{K} b_t(k) \sum_{l=1}^{L} \sigma(\pi_i^{(l)} \mid k) \cdot \pi_i^{(l)}(a_i \mid h_i). \tag{3}$$

This meta-policy formulation provides a principled way to combine prior agent policies in a belief-aware manner. Rather than committing to a single policy or performing hard switching, the agent combines its policies in a belief-weighted manner. This approach ensures robustness and adaptability, and is especially useful in planning contexts where simulated rollouts benefit from coherent and belief-aligned action selection.

**Payoff matrix estimation.** The meta-policy requires a payoff matrix $\mathbf{R} \in \mathbb{R}^{L \times K}$ whose entry $R_{l,k}$ is the empirical return of agent policy $\pi_i^{(l)}$ against latent opponent type $k$. To compute those entries we first label every offline trajectory with a type index and then aggregate the corresponding returns.

After the quantized autoencoder is trained, each collected opponent trajectory $\tau$ is passed through the encoder $f_{\text{enc}}$ to obtain an embedding $z$. The trajectory is mapped to its closest codebook vector $e_k$ by assigning the index $k^\star = \arg\min_k \|z - e_k\|_2^2$, which yields a latent label consistent with the prototype policies used later in belief tracking. Given the labels, the payoff matrix is estimated by bucket-wise averaging:

$$R_{l,k} = \frac{1}{N_{l,k}} \sum_{j:\, k_{l,j}^\star = k} G_{l,j}, \quad N_{l,k} = |\{j \mid k_{l,j}^\star = k\}|, \tag{4}$$

where $G_{l,j}$ is the return obtained by $\pi_i^{(l)}$ in trajectory $\tau_{l,j}$.

*Trajectory collection.* All trajectories used above are generated once offline by pairing a policy $\pi_i^{(l)}$ from the agent library $\Pi_i$ with a policy $\pi_{-i}^{(j)}$ from the opponent library $\Pi_{-i}$ and executing the environment model $\mathcal{G}$. Each trajectory simultaneously (i) supplies data for learning and labeling latent types and (ii) records the returns needed for the payoff matrix, so the same dataset serves both purposes without extra simulations. This payoff matrix construction is performed entirely offline, removing its computational cost from the online decision-making loop.

### 3.4 Planning with Quantized Opponent Models

Once we have established a compact latent representation of the opponent's policies through QOM, we integrate this model into a MCP framework for belief-aware planning. Our goal is to plan effective actions of the agent while reasoning over the uncertainty in the opponent's latent type. This integration requires three key steps: belief tracking, opponent simulation, and agent action selection.

**Algorithm 1** Planning with Quantized Opponent Models
_______________________________________________________________________________________
1: **Input:** initial history $h_i^{\text{real}}, h_{-i}^{\text{real}}$, belief $b^{\text{real}}(k)$
2: **for** each environment step **do**
3:     Initialize simulation with current history and belief
4:     **while** within computational budget **do**
5:       Set rollout history and belief from real environment values
6:       **while** not terminal and within depth limit **do**
7:         Sample opponent type $k \sim b(k)$
8:         Sample opponent action $a_{-i} \sim \tilde{\pi}_k(\cdot \mid h_{-i})$
9:         Compute agent meta-policy $\pi_i^{\text{meta}}$ and select agent action $a_i$ using PUCT
10:        Simulate next step: $(s', o_i', o_{-i}', r_i) \sim \mathcal{G}$
11:        Update histories: $h_i, h_{-i}$
12:        Update belief $b(k)$ using simulated opponent observation and Equation (5)
13:       **end while**
14:       Backpropagate value and update $Q, N$
15:     **end while**
16:     Select $a_i^t = \arg\max_{a_i} N(h_i, a_i)$
17:     Execute $a_i^t$, observe $o_i^{(t+1)}, a_{-i}^{(t)}$, infer $o_{-i}^{(t)}$ and update $h_i^{\text{real}}, h_{-i}^{\text{real}}$
18:     Update belief $b^{\text{real}}(k)$ using Equation (5), apply belief smoothing
19: **end for**
_______________________________________________________________________________________

**Belief tracking.** During both online execution and internal simulation, the agent maintains a categorical belief $b_t(k)$ over the $K$ latent types. After observing an opponent action $a_{-i}^{(t)}$ given history $h_{-i}^{(t)}$, either from real interaction or simulated rollout, the belief is updated via Bayes' rule:

$$b_{t+1}(k) \ \propto \ b_t(k) \left[ \tilde{\pi}_k(a_{-i}^{(t)} \mid h_{-i}^{(t)}) \right]^{\beta}, \tag{5}$$

Here, $\tilde{\pi}_k(a_{-i}^{(t)} \mid h_{-i}^{(t)})$ denotes the likelihood of the opponent action under latent type $k$, estimated by the learned decoder.

During online execution, belief updates are performed based on the true observed opponent actions, progressively refining the agent's uncertainty over latent types as interaction unfolds. This posterior refinement aligns with the assumptions underlying our theoretical analysis and supports the expected convergence behavior over time, see Section A.

During rollouts, belief updates are similarly performed at every step. At each newly expanded node, the belief is inherited from the parent node and immediately updated according to the newly generated opponent action. Formally, if a parent node at history $h_i^t$ carries belief $b_t(k)$, then after simulating the agent's action $a_i^{(t)}$ and opponent action $a_{-i}^{(t)}$, the newly created child node at $h_i^{t+1}$ carries an updated belief $b_{t+1}(k)$ according to Equation (5). This consistent inheritance and updating of belief across both real and hypothetical trajectories enables the agent to reason adaptively about latent opponent behavior over time.

To mitigate premature overconfidence caused by noisy observations or decoder approximation errors, we apply belief smoothing after each update by interpolating the updated belief $b_{t+1}(k)$ with the initial prior $b_0(k)$. Specifically, the smoothed belief is computed as $b_{t+1}^{\text{smooth}}(k) = (1 - \lambda)b_{t+1}(k) + \lambda b_0(k)$, where $\lambda$ controls the degree of smoothing and $b_0(k)$ is the uniform distribution over types. This simple smoothing mechanism helps maintain exploration and prevents belief collapse, especially during the early, data-sparse stages of interaction. The dynamically refined belief at each history serves as the foundation for both opponent simulation and meta-policy construction throughout planning and execution.

**Belief-driven simulation of opponent actions.** In each simulation, the agent leverages the current belief distribution $b_t(k)$ to model opponent behavior in a belief-aware manner. At each decision step during the rollout, a latent opponent type $k$ is sampled according to the current belief, $k \sim b_t(k)$. Given the sampled type, the corresponding decoder $\tilde{\pi}_k$ is used to simulate the opponent's action based on the reconstructed rollout history $h_{-i}^{(t)}$, following $a_{-i}^{(t)} \sim \tilde{\pi}_k(\cdot \mid h_{-i}^{(t)})$. Since the opponent's

private observations are not directly available, the agent reconstructs the opponent's history using the environment simulator $\mathcal{G}$ to infer plausible opponent observations during the deployment. The simulator generates next states and observations based on joint actions, enabling the agent to update $h_{-i}^{(t)}$ incrementally as simulation progresses.

This belief-driven simulation ensures that the planning evolves consistently with the agent's posterior over opponent types. As the belief $b_t(k)$ becomes increasingly concentrated through interaction, the sampled opponent behaviors naturally adapt, leading to progressively more targeted planning.

**Belief-informed agent action selection.** For selecting its own actions during simulation, the agent adopts a belief-informed variant of the Predictor Upper Confidence Bound for Trees (PUCT) algorithm [45] that incorporates the meta-policy as a dynamic prior. Specifically, for each candidate action $a_i$ at a given agent history $h_i$, the PUCT score is computed as:

$$ U(h_i, a_i) = Q(h_i, a_i) + c \cdot \frac{\pi_i^{\text{meta}}(a_i \mid h_i)}{1 + N(h_i, a_i)}, \tag{6} $$

where $Q(h_i, a_i)$ denotes the estimated action value of action $a_i$ in the agent history $h_i$, $N(h_i, a_i)$ is the corresponding visit count, $\pi_i^{\text{meta}}(a_i \mid h_i)$ represents the prior probability of selecting $a_i$ under the meta-policy constructed from the current belief, and $c > 0$ is an exploration constant. After each simulation, the value $Q(h_i, a_i)$ is updated by backing up the cumulative return $V$ along the search path, where $Q(h_i, a_i)$ is incrementally updated as a running average weighted by the new sample, and $V$ denotes the cumulative discount reward obtained after selecting action $a_i$ at history $h_i$. Specifically, $V$ is recursively computed during the rollout by discounting the future return and adding the immediate reward, following the standard form $V \leftarrow r_i + \gamma V$, where $r_i$ is the reward received and $\gamma \in [0, 1)$ is the discount factor.

By integrating $\pi_i^{\text{meta}}$ into the PUCT, the agent is guided towards actions that are promising given its belief about the opponent, while still allowing sufficient exploration of less visited branches. This mechanism ensures that search effort is adaptively focused on policies likely to perform well against the inferred opponent-type distribution, thereby improving planning efficiency.

Our method integrates belief tracking, opponent simulation, and meta-policy-guided action selection into the planning algorithm to enable adaptive opponent uncertainty. By reasoning about latent policy types and updating beliefs online, the agent can make progressively more informed decisions throughout planning and execution. The simplified planning procedure is shown in Algorithm 1, and the detailed algorithm is provided in Section B.

## 4 Experiments

### 4.1 Environment

We evaluate our method across four diverse multi-agent environments, covering a range of interaction structures, including cooperative and competitive dynamics, varying levels of partial observability, and a spectrum of strategic complexity.

The *Pursuit-Evasion* and *Predator-Prey* tasks [42] are discrete grid-world domains where agents act based on local observations and reactive strategies. Pursuit-Evasion is an asymmetric zero-sum game where the evader must reach a goal while avoiding capture by a pursuer. Predator-Prey is a cooperative setting requiring multiple predators to coordinate in order to capture the prey.

The *Running-with-Scissors (RWS)* environment [5, 53], extends rock-paper-scissors into a spatial, partially observable gridworld. Two agents collect resources—rock, paper, or scissors—and can tag a 3×3 area ahead to challenge the opponent, with outcomes determined by rock–paper–scissors rules. The environment features concealed information and non-transitive payoffs, requiring agents to infer opponent intent and commit to counter-strategies through temporally extended behavior.

The *One-on-One* scenario [23, 57] involves high-dimensional state and action spaces, where an attacker and a goal-keeper compete in a simplified football drill. This environment emphasizes fine-grained control and opponent-aware decision-making in high-dimensional state and action spaces.

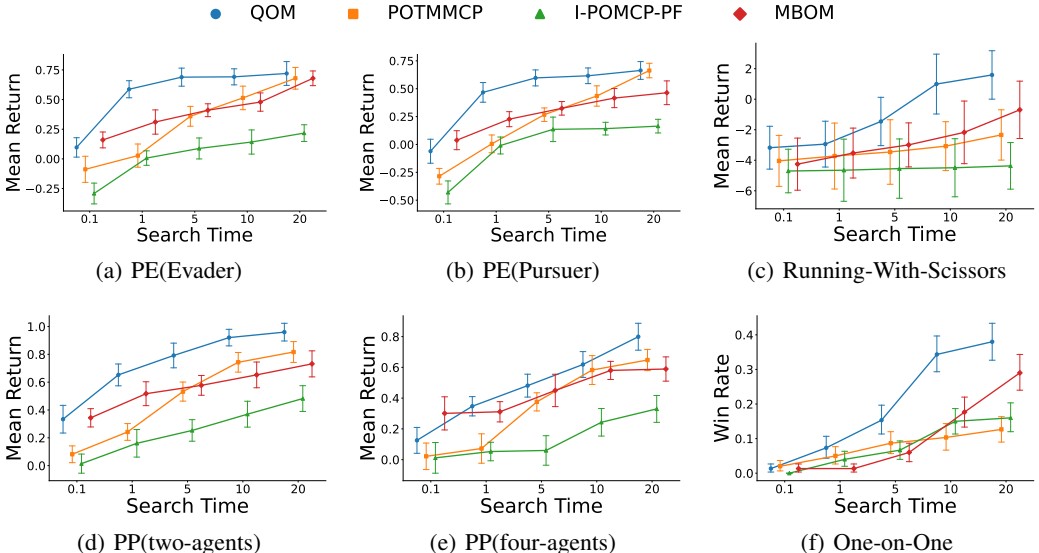

Figure 2: Performance comparison across six multi-agent environments under varying search time budgets. Each point denotes the mean return over 300 episodes, with error bars showing 95% confidence interval. Our method (QOM) consistently outperforms all baselines across a wide range of settings, particularly especially when computation time is limited. The x-axis corresponds to discrete search-time settings with non-uniform spacing in actual runtime; lines are drawn to illustrate performance trends. Environments include: (a)(b) Pursuit-Evasion from both evader and pursuer perspectives; (c) Running-With-Scissors; (d)(e) Predator-Prey with two and four agents; and (f) One-on-One football task.

## 4.2 Experimental Setup

**Policy Library.** To obtain a diverse set of strategies, we construct the policy library using PSRO [25]. Unless otherwise specified, each environment includes a library of $|\Pi_i| = 10$ agent policies and $|\Pi_{-i}| = 50$ opponent policies. The opponent's latent type size is set to $K = 16$ by default.

**Baselines.** We compare QOM against the following baselines that reflect different paradigms of belief-aware planning and opponent modeling.

- **POTMMCP** [41]. A state-of-the-art Monte-Carlo planning method that performs belief-aware decision making by maintaining a particle filter over opponent policies. It integrates belief tracking with search-time planning to adapt to opponent policy.
- **I-POMCP-PF.** A variant of I-POMDPs [19] without explicit communication or recursion. It uses particle filtering to represent beliefs over opponent policies and performs planning via POMCP with a uniform random rollout policy at the leaf nodes.
- **MBOM** [57]. A machine Theory of Mind method that infers opponent intentions through online planning and simulation rollouts. MBOM does not rely on a discrete policy library and instead models the opponent implicitly through predictive learning.

Each experiment is conducted over 300 episodes, and the average return is reported as the final result. We evaluate performance under varying search time budgets, scaled as $[0.1, 1, 5, 10, 20]$ units of time. The unit time is environment-specific and reflects the relative planning complexity of each domain. We use ground-truth simulators for all environments except One-on-One, which uses a learned simulator. Detailed experimental settings, including environment configurations, policy libraries, and hyperparameters, are provided in Section C.

## 4.3 Performance of static opponents

Figure 2 reports the mean return under varying search time budgets. Across all environments and time budgets, our proposed method QOM consistently achieves the highest performance. This indicates its

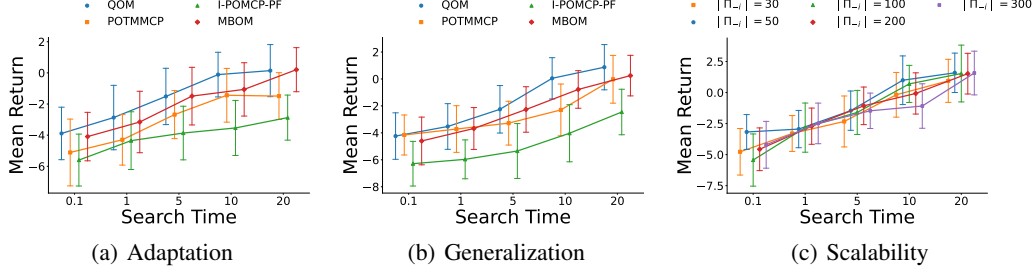

Figure 3: Evaluation in the Running-With-Scissors environment under three conditions. In (a), the opponent switches its policy after unsuccessfully tagging the agent, testing the agent's ability to rapidly adapt to changing behaviors. In (b), the opponent uses a policy not included in the predefined policy library, evaluating each method's generalization to unseen policies. In (c), we vary the policy library size $|\Pi_{-i}|$ to assess the scalability and inference efficiency of QOM as policy library size increases.

ability to efficiently leverage posterior beliefs over opponent types to guide decision making, even under tight computational constraints. In particular, QOM shows rapid improvement with increased search time in both the Pursuit-Evasion tasks and the Predator-Prey, demonstrating effective belief-driven planning in both asymmetric and cooperative-competitive scenarios. POTMMCP employs particle-filtered posterior tracking to model opponent behavior. However, under low search time budgets, its effectiveness diminishes due to the sample inefficiency of particle filtering. In contrast with QOM's compact type-based belief model, which enables faster and more structured inference. I-POMCP-PF shows moderate performance gains but remains limited by its reliance on uniform rollout policies and shallow belief updates, especially in complex environments such as Figure 2(e). Meanwhile, MBOM exhibits competitive performance in simpler tasks but degrades in more strategic domains as Figure 2(c), because it lacks an explicit policy library and struggles to generalize online. In the One-on-One setting as Figure 2(f), the performance variance is higher across all methods due to the fine-grained strategic adaptations required. Even here, QOM still maintains a slight advantage.

## 4.4 Adaptation to Switching Opponents

To evaluate adaptability, we design an experiment where the opponent switches its policy after unsuccessfully tagging the agent. This simulates a piecewise-stationary adversary, requiring the agent to detect behavioral changes and respond accordingly. As shown in Figure 3(a), QOM outperforms all baselines. This highlights the efficiency of our Bayesian belief update mechanism in detecting type shifts and reallocating planning effort. In contrast, POTMMCP and I-POMCP-PF struggle to adapt rapidly due to their reliance on particle filtering, which lags in capturing abrupt type transitions with limited samples. MBOM performs comparably at high budgets but lacks structured uncertainty modeling, resulting in less reliable performance under tight time constraints.

## 4.5 Generalization to Unseen Opponents

We test generalization by deploying an opponent that uses a policy not included in the policy library. This scenario evaluates whether the agent can perform robustly when faced with out-of-distribution behaviors. As shown in Figure 3(b), QOM again demonstrates strong performance, outperforming baselines across all search budgets. Despite the unseen nature of the opponent, the quantized opponent models extrapolate effectively, assigning probability mass to nearby types in latent space.

## 4.6 Scalability with Policy Library Size

To assess the scalability of belief inference, we vary the policy library size $|\Pi_{-i}|$ from 30 to 300 while keeping the test policy fixed. As shown in Figure 3(c), QOM maintains stable and high performance across all library sizes, showing only minimal degradation, highlighting the scalability of the belief model. This demonstrates the model's ability to scale efficiently with growing policy library size. In contrast, particle-based methods such as POTMMCP suffer from increasing posterior variance

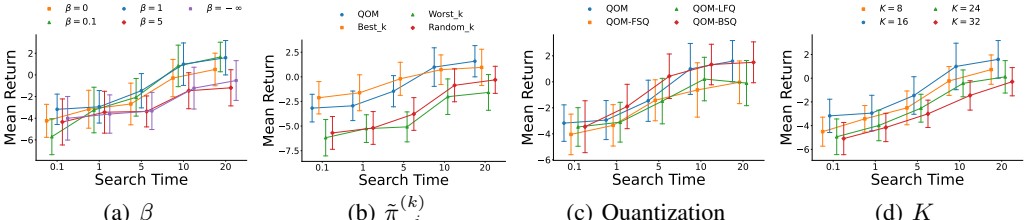

(a) $\beta$      (b) $\tilde{\pi}_{-i}^{(k)}$      (c) Quantization      (d) $K$

Figure 4: Ablation study in the Running-With-Scissors environment. In (a), we vary the belief update temperature $\beta$, observing that moderate entropy regularization improves adaptation, while extremes degrade performance. In (b), we compare different approximations of the belief-weighted opponent policy $\tilde{\pi}_{-i}^{(k)}$, showing that marginalizing over type beliefs (QOM) outperforms heuristic alternatives such as choosing the best, worst, or random type. In (c), we evaluate different quantization methods used to discretize opponent behavior, including FSQ, LFQ, and BSQ; all perform comparably, but VQ-based QOM achieves more consistent gains across budgets. In (d), increasing the number of types $K$ initially improves performance, but further increases lead to *type* redundancy.

and degraded precision when $|\Pi_{-i}|$ is large, due to the curse of dimensionality in sampling-based inference.

## 4.7 Ablation Study

**Effect of Belief Temperature $\beta$.** We examine the impact of the entropy temperature $\beta$ in the belief update, which controls the sharpness of the posterior over opponent types. As shown in Figure 4(a), moderate values (e.g., $\beta = 1$) yield the best performance, striking a balance between confident updates and robustness to uncertainty. When $\beta = 0$ (uniform), the model performs hard inference, which is brittle in early episodes. Conversely, $\beta \to \infty$ (greedy) leads to confident planning, resulting in degraded performance. This result confirms the importance of uncertainty calibration in belief-aware planning.

**Opponent Simulation policy $\tilde{\pi}_{-i}^{(k)}$.** To validate the design choice of using a belief-weighted mixture of opponent types during simulation, we compare against three ablations: choosing the most likely type (Best-$k$), the least likely (Worst-$k$), and a random type. Note that Best-$k$ and Worst-$k$ are not realizable strategies but post-hoc performance bounds, obtained by selecting the best or worst single opponent type in hindsight. As shown in Figure Figure 4(b), the QOM agent using soft marginalization consistently outperforms all alternatives. Heuristic approximations either inject high variance (Random) or suffer from brittle assumptions, especially under limited data. This highlights the benefit of maintaining structured uncertainty over opponent behavior throughout planning.

**Quantization Method.** We evaluate the influence of different quantization schemes used to discretize the latent opponent policy space: FSQ [33], LFQ [56], and BSQ [58]. Figure 4(c) shows that all methods yield comparable overall performance, but VQ-based QOM achieves slightly more stable results across search budgets. While structured quantization variants may offer theoretical benefits, the core advantage stems from converting continuous trajectory embeddings into discrete types that enable Bayesian reasoning. This suggests the proposed framework is flexible to various encoder discretization strategies.

**Number of Opponent Types $K$.** We study how the number of discrete opponent types $K$ affects performance. As shown in Figure 4(d), increasing $K$ initially improves return, as a larger type enables finer distinctions among opponent behaviors. However, beyond a certain point, additional types provide diminishing returns and may lead to type redundancy, where multiple types generate similar behaviors. This can increase the variance of belief updates and reduce planning efficiency. The result suggests that while increasing $K$ allows modeling more diverse behaviors, an excessive number of types leads to small distances between them, making the belief less informative and weakening generalization. For more results from other environments, see Section D.

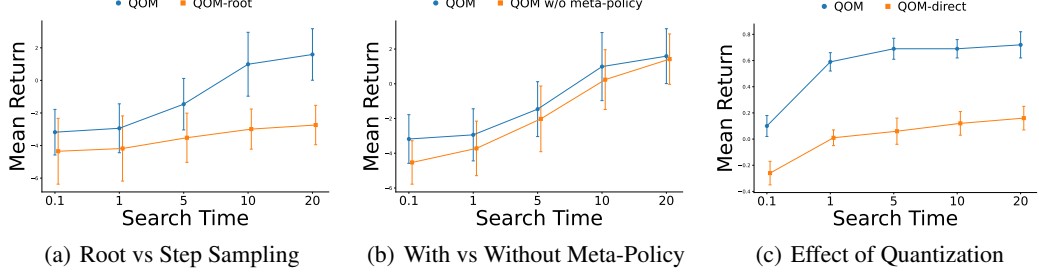

|  (a) Root vs Step Sampling | (b) With vs Without Meta-Policy | (c) Effect of Quantization |

Figure 5: Ablation studies of QOM variants. (a) illustrates that dynamic opponent sampling with belief updates at each step enables faster adaptation and higher returns in the Running-With-Scissors. (b) indicates that the meta-policy substantially improves planning efficiency in the Running-With-Scissors. (c) shows that QOM achieves better performance under tight search budgets owing to efficient belief updates and type compression in the Pursuit-Evasion.

**Root vs. Step-Level Opponent Sampling** To examine the effect of dynamic belief refinement, we compare QOM-step (our default), which re-samples opponent types and updates beliefs at each simulation step, with QOM-root, which samples only once at the root and fixes the opponent type throughout the rollout. While the latter approximates marginalization over types, it fails to capture evolving posterior beliefs as simulated trajectories unfold. As shown in Figure 5(a), QOM-step achieves consistently higher returns across all search budgets, indicating that dynamic belief updates provide a clear advantage by maintaining uncertainty over latent opponent types during planning.

**With vs. Without Meta-Policy** We further assess the role of the meta-policy used during simulation. The idea of a meta-policy originates from POTMMCP; our method extends it by integrating a learned quantized opponent model. To isolate this component, we compare the standard QOM (with meta-policy) against a variant that uses uniform rollouts (QOM w/o meta-policy). As shown in Figure 5(b), removing the meta-policy consistently degrades performance, confirming that belief-aware rollout selection enhances planning.

**Effect of Opponent Quantization** We include QOM-direct, which performs exact categorical Bayesian updates over all 50 opponent policies without quantization, using the same PUCT prior, belief smoothing, and history reconstruction. As shown in Figure 5(c), under low computational budgets, QOM outperforms QOM-direct, while their performance converges at higher budgets. This pattern highlights that the quantized representation accelerates belief updates and focuses exploration, whereas the unquantized variant incurs costly marginalization and slower posterior concentration when policy types are not easily separable.

# 5 Conclusion and Limitation

We introduced the *Quantized Opponent Models* (QOM), a framework that combines type-based opponent modeling with Monte-Carlo Planning. By learning a discrete set of opponent types and maintaining a Bayesian belief online, QOM enables efficient and adaptive planning under uncertainty. Experiments show consistent gains over strong baselines across multiple environments and settings.

However, QOM relies on a fixed policy library and assumes short-term stationarity of opponent policies. Its performance may degrade if opponent policies lie far outside the learned type space. Future work could explore continual type discovery and hierarchical dynamics settings.

# Acknowledgments

This work was supported by NSFC under Grant 62450001 and 62476008. The authors would like to thank the anonymous reviewers for their valuable comments and advice.

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

# A  Theoretical Analysis

In this section, we provide theoretical guarantees for the proposed QOM method, establishing how the autoencoder-based quantization affects learning performance. We follow the standard analysis structure with notation and assumptions, then present the key lemmas and theorems. Throughout, we explicitly highlight the influence of the quantization error on posterior concentration.

**Notation.** Let $\mu^*$ denote the true underlying distribution or model of the environment. In the context of QOM, $\mu^*$ can be considered the true (unknown) distribution over high-dimensional observations. The autoencoder compresses observations into a finite codebook of $K$ prototypical points (or latent states); we denote this approximate model distribution by $\hat{\mu}$. The distribution $\hat{\mu}$ places probability mass $\tilde{\pi}_k$ on the $k$-th code (for $k = 1, \ldots, K$), which aims to approximate the true mass $\pi_k$ that $\mu^*$ assigns to the region represented by that code. We write $p_\mu(x)$ for the probability density (or mass) of observation $x$ under a distribution $\mu$. Notation $\mathrm{KL}(\mu^* \| \mu)$ denotes the Kullback–Leibler (KL) divergence of $\mu$ from $\mu^*$.

Corresponding to the discussion in Section 3.4, the distribution $\mu$ can be seen as the belief $b_\mu(k)$ given a fixed opponent pool $\{\pi_k\}_{k=1}^K$. The observation $x$ represents the opponents' history $x = (h_{-i}, a_{-i})$. Therefore, the probability $p_\mu(x)$ can be formulated as

$$p_\mu(x) = \sum_k b_\mu(k)\pi_k(a_{-i}|h_{-i}). \tag{7}$$

$\mu^*$ corresponds to the unknown ground truth belief $b_{\mu^*}(k)$

The QOM agent maintains a Bayesian posterior $\Pr(\mu|X_{1:t})$ over models $\mu$ given the data $X_{1:t}$ up to time $t$. We define $\phi(T) := \log T \log \log T$, a slowly growing function of the horizon that will appear as a correction factor due to temporal dependence. For clarity, $|\cdot|$ denotes an $\ell_2$ norm unless stated otherwise.

**Assumptions.** We assume the following conditions hold in the QOM model setting:

*Assumption 1 (Boundedness and $\beta$-mixing).* The sequence of observations (or contexts) $\{X_t\}_{t=1}^T$ is generated from a $\beta$-mixing process with mixing coefficients decaying sufficiently fast (e.g. $\beta(m) = O(e^{-cm})$ for some $c > 0$), ensuring weak dependence across time. Moreover, all relevant random variables are bounded: in particular, there exists $B > 0$ such that for all $t$ and for all candidate models $\mu$, the log-likelihood ratio per observation is bounded by $B$:

$$\left| \log \frac{p_\mu(X_t)}{p_{\mu^*}(X_t)} \right| \le B, \qquad \text{almost surely.} \tag{8}$$

This implies the rewards or losses are also bounded (for instance, if rewards $r_t \in [0, B]$ or the maximum reward gap is bounded by $B$). These conditions allow us to apply Bernstein-type concentration inequalities even though data are dependent over time.

*Assumption 2 (Autoencoder multiplicative approximation).* The autoencoder provides a *multiplicative reconstruction guarantee*: there exists a fixed $\xi \in (0, 1)$ (the quantization error parameter) such that for every observation $X_t$ and its autoencoder reconstruction $\tilde{X}_t$, the relative error is uniformly bounded:

$$\|X_t - \tilde{X}_t\| \le \xi \|X_t\|, \qquad \forall\, t = 1, \ldots, T. \tag{9}$$

In words, the reconstruction is accurate up to a factor $\xi$ of the original magnitude in every instance. This assumption implies that the approximate distribution $\hat{\mu}$ induced by the codebook is close to the true distribution $\mu^*$ in a multiplicative sense. For example, we can reasonably assume (perhaps by a Lipschitz continuity argument on $p_\mu(x)$) that for all $x$,

$$(1 - \xi)\, p_{\mu^*}(x) \le p_{\hat{\mu}}(x) \le (1 + \xi)\, p_{\mu^*}(x). \tag{10}$$

That is, the density under the QOM model does not deviate from the true density by more than a $\pm\xi$ relative factor at any point. Finally, we assume the true model $\mu^*$ (or a distribution arbitrarily close to it in KL) lies within the support of our Bayesian prior – formally, the prior assigns sufficient mass to a small KL-neighborhood of $\mu^*$. This ensures the posterior is well-behaved around the truth.

Under these assumptions, we now derive step-by-step the main theoretical results: a bound on the KL divergence due to quantization (Theorem 1), a posterior concentration rate (Theorem 2). Each result explicitly incorporates the $\beta$-mixing correction $\phi(T)$ and the quantization error $\xi$.

**Lemma 1** (KL divergence under quantization). *Under Assumption 2, the KL divergence between the true distribution $\mu^*$ and the autoencoder-quantized model $\hat{\mu}$ is bounded by a small constant on the order of $\xi^2$. In particular, one may show*

$$\mathrm{KL}(\mu^*\|\hat{\mu}) \ \leq \ -\log(1-\xi) \approx \xi + \tfrac{\xi^2}{2} + O(\xi^3) \ < \ \frac{\xi}{1-\xi}.$$

*Consequently, for sufficiently small $\xi$, $\mathrm{KL}(\mu^*\|\hat{\mu})$ is $O(\xi)$ (and in particular $O(\xi^2)$ to leading quadratic order).*

*Proof.* Since $p_{\hat{\mu}}(x)$ everywhere pointwise approximates $p_{\mu^*}(x)$ to within a $(1 \pm \xi)$ factor, the KL divergence can be bounded by the worst-case log discrepancy. By definition,

$$\mathrm{KL}(\mu^*\|\hat{\mu}) \ = \ \int_{\mathcal{X}} p_{\mu^*}(x) \log \frac{p_{\mu^*}(x)}{p_{\hat{\mu}}(x)} \, dx \ . \tag{11}$$

Given the multiplicative bounds, we have $\frac{p_{\mu^*}(x)}{p_{\hat{\mu}}(x)} \leq \frac{1}{1-\xi}$ for all $x$. Thus $\log \frac{p_{\mu^*}}{p_{\hat{\mu}}} \leq \log \frac{1}{1-\xi}$. Furthermore, on the other side $p_{\hat{\mu}}(x) \leq (1+\xi)p_{\mu^*}(x)$ implies $\frac{p_{\mu^*}}{p_{\hat{\mu}}} \geq \frac{1}{1+\xi}$, so $\log \frac{p_{\mu^*}}{p_{\hat{\mu}}} \geq \log \frac{1}{1+\xi} = -\log(1+\xi)$. Since $-\log(1+\xi) < -\log(1-\xi)$ for $0 < \xi < 1$, the maximal absolute divergence occurs when $p_{\hat{\mu}}(x)$ is smaller than $p_{\mu^*}(x)$ by factor $(1-\xi)$. In the worst case (if this held for all $x$), $\mathrm{KL}(\mu^*\|\hat{\mu}) = \log \frac{1}{1-\xi}$. In reality $p_{\hat{\mu}}$ will not be uniformly at the extreme bound for all $x$, so $\mathrm{KL}(\mu^*\|\hat{\mu})$ will be strictly less than this worst-case. In any case, $\log \frac{1}{1-\xi}$ expands to $\xi + \frac{\xi^2}{2} + \frac{\xi^3}{3} + \cdots < \xi/(1-\xi)$, giving the stated bound. Thus the divergence between the true distribution and the QOM's approximate distribution is on the order of the small quantization error $\xi$, as claimed. $\square$

Intuitively, Lemma 1 shows that the autoencoder-based quantization does not significantly distort the problem from an information-theoretic perspective: the true environment $\mu^*$ is almost as likely under the quantized model class as it is under the truth (their KL divergence is very small for small $\xi$). This fact will be crucial in proving posterior concentration, since it guarantees that the true (or near-true) model is not "ruled out" by a large divergence.

**Theorem 2** (Posterior Concentration under QOM). *Under Assumptions 1 and 2, the Bayesian posterior concentrates around the true model $\mu^*$ at a rate accounting for the dependence and quantization. In particular, there exist positive constants $C_1, C_2$ such that for sufficiently large $T$, with high probability (approaching 1 as $T \to \infty$):*

$$\Pr\Big(\mu : \ \mathrm{KL}(\mu^*\|\mu) > C_1 \, \frac{\phi(T)}{T} \ \Big| \ X_{1:t}\Big) \ \leq \ \exp\Big(-C_2 \, \frac{T}{\phi(T)}\Big),$$

*up to lower-order terms. Equivalently, the posterior places almost all its mass on those models $\mu$ that satisfy $\mathrm{KL}(\mu^*\|\mu) = O\left(\frac{\log T \log \log T}{T}\right)$. In particular, the bulk of the posterior mass lies within an $O\left(\frac{\log T \log \log T}{T}\right)$ KL-neighborhood of $\mu^*$. Combined with Lemma 1, this implies that the posterior is effectively concentrated near the true $\mu^*$ (or an equivalent $\hat{\mu}$ that is $\xi$-close to $\mu^*$) up to an asymptotic approximation error on the order of $\xi$.*

*Proof.* The proof is based on a Bernstein-type large deviation inequality for $\beta$-mixing sequences, which allows us to handle the dependent observations. At a high level, we need to show that for any candidate model $\mu$ that is significantly different from $\mu^*$ (say, $\mathrm{KL}(\mu^*\|\mu)$ is larger than some $\varepsilon$), the posterior probability of $\mu$ becomes negligibly small as $T$ grows. By Bayes' rule,

$$\Pr(\mu \mid X_{1:t}) \ \propto \ \Pr(\mu) \prod_{t=1}^{T} \frac{p_\mu(X_t)}{p_{\mu^*}(X_t)} \, ,$$

where $\Pr(\mu)$ in the numerator is the prior density at $\mu$. Consider any model $\mu$ with $\mathrm{KL}(\mu^*\|\mu) = \eta > 0$. Define the log-likelihood ratio sum:

$$L_T(\mu) := \sum_{t=1}^{T} \log \frac{p_\mu(X_t)}{p_{\mu^*}(X_t)} .$$

The expected value of this sum under the true distribution is negative and scales with $T$:

$$
\begin{aligned}
\mathbb{E}_{\mu^*}[L_T(\mu)] &= \sum_{t=1}^{T} \mathbb{E}_{\mu^*}\left[\log \frac{p_\mu(X_t)}{p_{\mu^*}(X_t)}\right] \\
&= -T \cdot \mathrm{KL}(\mu^*\|\mu) \quad \text{(from (11) and Assumption 1 that } \{X_t\} \text{ is a } \beta\text{-mixing process)} \\
&= -T\eta
\end{aligned}
$$

$$(12)$$

We want to show that $L_T(\mu)$ is very likely to be much less than $0$ (i.e. strongly negative), so that $\prod_{t=1}^{T} \frac{p_\mu(X_t)}{p_{\mu^*}(X_t)} = \exp(L_T(\mu))$ is extremely small, overcoming even a nonzero prior mass on $\mu$. To this end, we apply a concentration inequality for $\beta$-mixing processes. Using a result of [22] on large deviations for $\beta$-mixing time series (a Bernstein-type inequality accounting for dependence), one can show that for any $\delta \in (0,1)$:

$$\Pr_{\mu^*}\left(L_T(\mu) > -\tfrac{1}{2}T\eta\right) \leq \exp\left(-C\,\frac{T\eta}{\phi(T)}\right),$$

for some constant $C > 0$ depending on the mixing parameters (including $v_\beta$) and the bound $B$. In words, with probability at least $1 - \exp(-CT\eta/\phi(T))$, we have $L_T(\mu) < -\tfrac{1}{2}T\eta$, meaning the log-likelihood ratio for $\mu$ is strongly negative (at least half of its negative expectation). When this event holds, the posterior weight of $\mu$ is exponentially small:

$$\Pr(\mu \mid X_{1:t}) \propto \Pr(\mu)\exp(L_T(\mu)) < \Pr(\mu)\exp(-\tfrac{1}{2}T\eta) .$$

Now, provided the prior $\Pr(\mu)$ is not pathologically large (which we can ensure since the prior is fixed and $\eta$ is a constant gap), the right-hand side decays super-polynomially in $T$. We can union-bound this argument over a suitable $\varepsilon$-net covering the set of all models with $\mathrm{KL}(\mu^*\|\mu) > \varepsilon$ to conclude that, with high probability, *no* such model has substantial posterior mass. More formally, for any $\varepsilon > 0$:

$$\Pr_{\mu^*}\left(\sum_{\{\mu:\mathrm{KL}(\mu^*\|\mu)>\varepsilon\}} \Pr(\mu \mid X_{1:t}) > \delta\right) \to 0 \qquad (T \to \infty),$$

for arbitrary small $\delta > 0$. This implies that asymptotically, $\Pr(\mathrm{KL}(\mu^*\|\mu) > \varepsilon \mid X_{1:t})$ converges to $0$. By choosing $\varepsilon$ on the order of $C_1 \frac{\phi(T)}{T}$ (the "critical radius" where the exponential bound starts to kick in, balancing $T\eta$ against $\phi(T)$ in the exponent above), we obtain the stated rate of posterior contraction. In summary, the posterior concentrates in a KL-ball around $\mu^*$ of radius $O(\frac{\log T \log\log T}{T})$. Finally, since $\mu^*$ is within $\xi$ (KL-)distance of some model in our support (by Lemma 1, $\hat{\mu}$ satisfies $\mathrm{KL}(\mu^*\|\hat{\mu}) = O(\xi)$), the posterior effectively centers around the $\hat{\mu}$ that approximates $\mu^*$, which in turn is $\xi$-close to $\mu^*$. Thus the presence of the quantization error $\xi$ does not prevent concentration; it only imposes that the posterior cannot distinguish models closer than $O(\xi)$ to $\mu^*$, thereby introducing a small $O(\xi)$ uncertainty floor even as $T \to \infty$. $\qquad\square$

Theorem 2 establishes that QOM's Bayesian posterior *rapidly concentrates* around the true environment's distribution, up to the minor resolution limit imposed by quantization. The $\beta$-mixing condition causes a logarithmic slow-down (the $\phi(T) = \log T \log\log T$ factor) compared to the iid case, but this is a very mild penalty. Importantly, the autoencoder's approximation error $\xi$ enters only as an additive term in the final concentration radius, meaning that aside from an $O(\xi)$ neighborhood that

the posterior cannot distinguish, the uncertainty shrinks at the usual $\tilde{O}(1/T)$ rate (up to log factors). In practical terms, as long as $\xi$ is small, the posterior quickly hones in on a model that is essentially as good as the truth for decision-making purposes.

## B   Algorithm

---

**Algorithm 2** Planning with Quantized Opponent Models

---

1: **Input:** initial agent history $h_i^{\text{real}}$, opponent history $h_{-i}^{\text{real}}$, belief $b^{\text{real}}(k)$
2: **for** each real environment step $t = 0, 1, 2, \ldots$ **do**
3:     *//Run simulations for action planning*
4:     **while** within computational budget **do**
5:         Initialize rollout history $h_i \leftarrow h_i^{\text{real}}$, $h_{-i} \leftarrow h_{-i}^{\text{real}}$
6:         Initialize rollout belief $b(k) \leftarrow b^{\text{real}}(k)$
7:         **while** node at $h_i$ is not terminal and depth has not exceeded maximum limit **do**
8:             **if** node at $h_i$ is not yet expanded **then**
9:                 Expand new child nodes at $h_i$ for each legal agent action
10:                 Initialize visit counts $N(h_i, a_i) = 0$ and action-values $Q(h_i, a_i) = 0$
11:             **end if**
12:             Sample opponent latent type $k \sim b(k)$
13:             Sample opponent action $a_{-i} \sim \tilde{\pi}_k(\cdot \mid h_{-i})$
14:             Compute meta-policy prior $\pi_i^{\text{meta}}(\cdot \mid h_i)$
15:             Compute PUCT scores as Equation (6) for all $a_i$
16:             Select agent action: $a_i = \arg\max_{a_i} U(h_i, a_i)$
17:             Append $(h_i, a_i)$ to search path
18:             Step environment simulator: $(s', o_i', o_{-i}', r_i) \sim \mathcal{G}(s, a_i, a_{-i})$
19:             Update rollout history: $h_i \leftarrow h_i \cup (o_i', a_i)$, $h_{-i} \leftarrow h_{-i} \cup (o_{-i}', a_{-i})$
20:             Update belief $b(k)$ by Equation (5)
21:         **end while**
22:         *//Perform rollout evaluation and update planning statistics*
23:         **for** each $(h_i, a_i)$ in the search path (in reverse order) **do**
24:

$$Q(h_i, a_i) \leftarrow \frac{Q(h_i, a_i) \times N(h_i, a_i) + V}{N(h_i, a_i) + 1}$$
$$N(h_i, a_i) \leftarrow N(h_i, a_i) + 1$$

25:             Update cumulative return: $V \leftarrow r_i + \gamma \times V$
26:         **end for**
27:     **end while**
28:     Select final action to execute: $a_i^t = \arg\max_{a_i} N(h_i, a_i)$
29:     *//Execute selected action and update real histories and beliefs*
30:     Execute $a_i^t$ in the real environment
31:     Observe agent observation $o_i^{(t+1)}$ and opponent action $a_{-i}^{(t)}$
32:     Infer opponent observation $o_{-i}^{(t)}$ by $\mathcal{G}$
33:     Update history: $h_i^{\text{real}} \leftarrow h_i^{\text{real}} \cup (o_i^{(t+1)}, a_i^t)$, $h_{-i}^{\text{real}} \leftarrow h_{-i}^{\text{real}} \cup (o_{-i}^{(t)}, a_{-i}^{(t)})$
34:     Update real-world belief $b^{\text{real}}(k)$ as Equation (5) and apply belief smoothing
35: **end for**

---

## C  Experiments Settings

### C.1  Environments

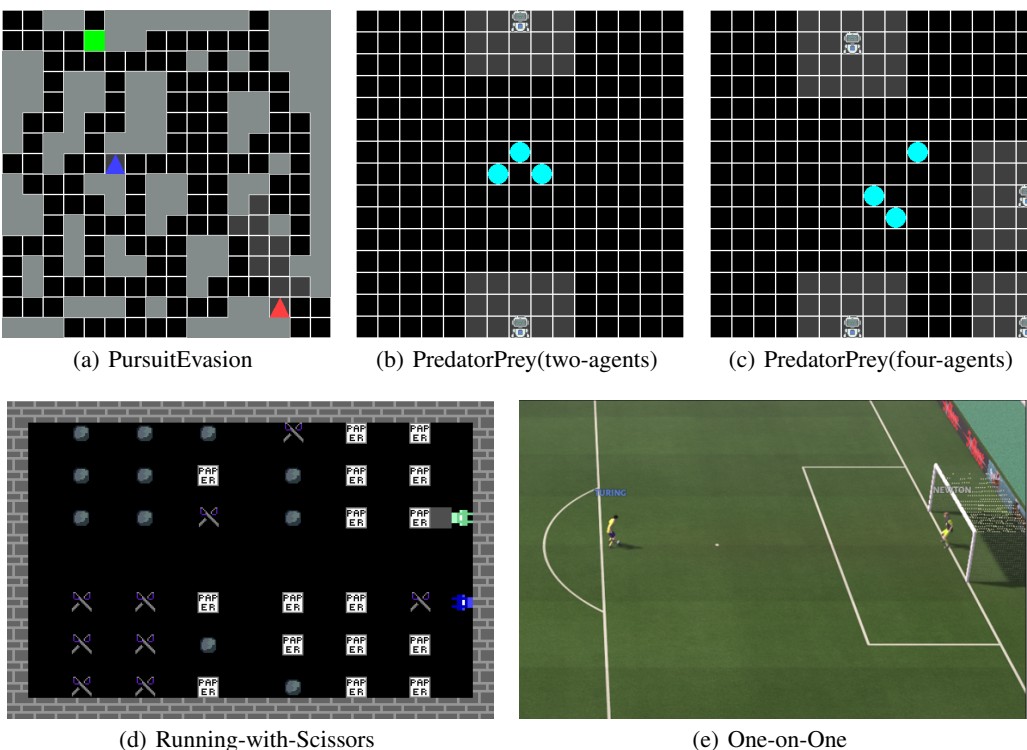

| (a) PursuitEvasion | (b) PredatorPrey(two-agents) | (c) PredatorPrey(four-agents) |

| (d) Running-with-Scissors | (e) One-on-One |

Figure 6: Environments

To evaluate our method across a broad range of multi-agent scenarios, we consider four partially observable environments spanning gridworlds, game-theoretic interactions, and high-dimensional control. All environments are sourced from established open-source platforms, including POSG-Gym [42], DeepMind Lab2D [5], and Google Research Football [23]. Figure 6 shows example visualizations.

**Pursuit-Evasion.** This environment features two agents—a pursuer and an evader—navigating a maze-like gridworld with limited observability. At each timestep, agents receive localized and noisy observations: wall proximity in four directions, directional visual cues, and auditory signals within a two-cell radius. The evader earns small rewards for advancing toward a safe zone, while the pursuer is rewarded for capturing the evader and penalized for failing to hinder progress. An episode ends when the evader escapes, is captured, or 100 steps elapse.

**Predator-Prey.** We consider two variants: one with two predators and another with three. Agents operate in a gridworld with deterministic dynamics and local $5 \times 5$ observations. Each episode includes three prey, and predators must coordinate to capture them. In the two-agent case, both predators must occupy the prey's cell; in the three-predator variant, at least three agents are required. Episodes conclude when all prey are caught or after 50 steps. Success hinges on coordination and inferring prey intentions under partial observability.

**Running-with-Scissors.** This environment extends the classic rock-paper-scissors game into a spatial, time-extended setting. Agents navigate a $13 \times 21$ grid containing 36 resource tiles—rock, paper, or scissors—half of which are randomized each episode. Agents observe a $4 \times 4$ local region and collect resources by stepping on tiles. Each starts with one unit of each resource. Agents can "mark" a $3 \times 3$ zone in front of them to challenge opponents, resolving confrontations via standard

RPS rules. Episodes last up to 500 steps or end upon a successful mark. Partial observability, randomized resources, and delayed interaction create a rich strategic environment requiring agents to infer and adapt to opponents' evolving inventories.

**One-on-One.** In this soccer-inspired setting, an attacker and a goalkeeper face off in a duel. The attacker controls the ball and may dribble or shoot at any time, while the goalkeeper must intercept. The attacker earns $+1$ for scoring; otherwise, the reward goes to the goalkeeper. We control the goalkeeper, treating the attacker as a non-stationary opponent. The continuous state-action space, visual inputs, and stochastic dynamics pose a significant challenge for belief-aware planning under opponent uncertainty.

## C.2 Policy Libraries and Payoff Matrix

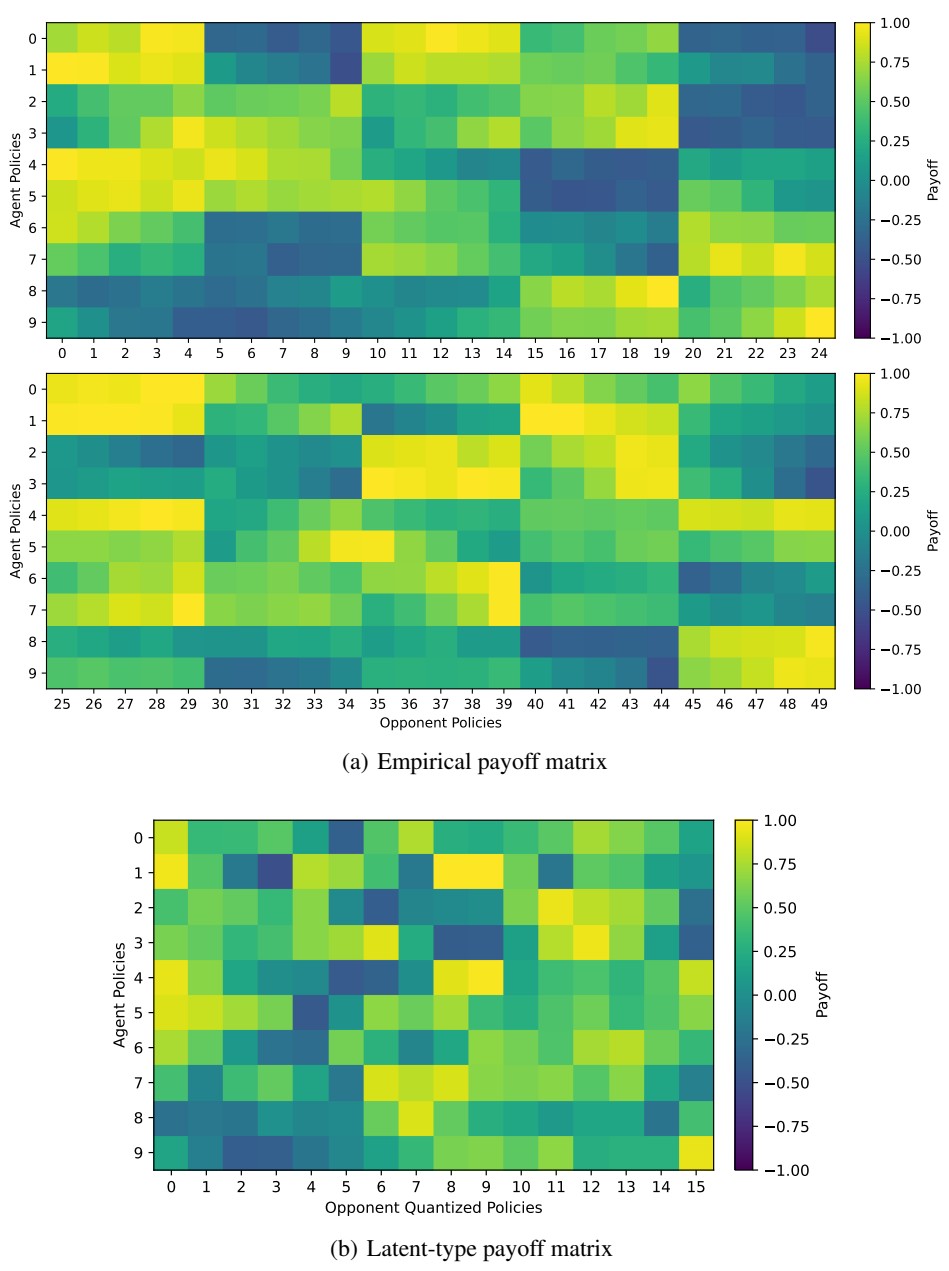

(a) Empirical payoff matrix

(b) Latent-type payoff matrix

Figure 7: Empirical payoff matrix and latent-type payoff matrix in Running-with-Scissors.

**Building the policy libraries.** Before planning begins, we generate both an *agent policy library* and an *opponent policy library* with Policy-Space Response Oracles (PSRO) [25, 24]. PSRO proceeds in rounds: at every iteration it trains a best-response policy to the current mixture of opponents and then adds this new policy to the empirical game. After PSRO iterations we have 10 distinct agent policies and 10 corresponding opponent polices. All oracles are trained with Proximal Policy Optimisation (PPO) [40], using the same network architecture and hyper-parameters reported in Section C.3. To widen behavioral coverage we further enlarge the opponent library with intermediate PPO checkpoints saved during training. Including these extra snapshots yields fifty opponent policies in total, spanning a broad spectrum of play styles from early exploration to converged behavior.

**Constructing the payoff matrix.** Running PSRO already produces an empirical payoff table that records the return obtained by each agent policy against every opponent policy. When Quantized Opponent Models (QOM) later need a payoff matrix indexed by latent opponent types, we simply *reuse* these existing simulation results, as Equation (4). Each opponent trajectory is labelled with a latent type by the quantized auto-encoder; returns from trajectories that share the same label are averaged, giving a payoff entry for that latent type. This recycling step eliminates the need for any additional rollouts while keeping the statistical link between the PSRO game and the latent-type abstraction used by QOM. We plot the payoff matrix in Running-with-Scissors, as shown in Figure 7.

**Unseen Opponent Policies.** For the generalization experiments, the unseen opponent policies are generated from three additional rounds of PSRO training, independent of those used to construct the QOM training library. These extra policies are excluded from the quantized opponent model and introduce novel strategic behaviors arising from different PSRO dynamics across runs. This setup enables evaluation of QOM's generalization to previously unseen opponent strategies beyond the fixed policy catalog.

## C.3 Hyper-parameters

Table 1: Hyperparameters

| | **Parameters** | Pursuit-Evasion | Predator-Prey | RWS | One-on-One |
|---|---|---|---|---|---|
| PPO | hidden units | MLP[64, 32] | MLP[64, 32] | MLP[64, 32] | MLP[64, 32] |
| | activation function | ReLU | ReLU | ReLU | ReLU |
| | optimizer | Adam | Adam | Adam | Adam |
| | learning rate | 0.0005 | 0.0005 | 0.0005 | 0.001 |
| | target update interval | 10 | 10 | 10 | 10 |
| | value discount factor | 0.99 | 0.99 | 0.99 | 0.99 |
| | GAE parameter | 0.99 | 0.99 | 0.99 | 0.99 |
| | clip parameter | 0.115 | 0.115 | 0.115 | 0.115 |
| | max grad norm | 0.5 | 0.5 | 0.5 | 0.5 |
| Policy Set | method | PSRO | PSRO | PSRO | Paired Policy Archive |
| | meta strategy method | $\alpha$-rank | $\alpha$-rank | $\alpha$-rank | |
| | sims per entry | 100 | 100 | 100 | |
| | gpsro iterations | 10 | 10 | 10 | |
| QOM | $\mid \Pi_i \mid$ | 10 | 10 | 10 | 10 |
| | $\mid \Pi_{-i} \mid$ | 50 | 50 | 50 | 50 |
| | learning rate | 0.0001 | 0.0001 | 0.0001 | 0.0001 |
| | reconstruction weight | 1 | 1 | 1 | 1 |
| | batch size | 64 | 64 | 64 | 64 |
| | letent types $K$ | 16 | 16 | 16 | 16 |
| | temperature $\beta$ | 1 | 1 | 1 | 1 |
| | exploration constant $c$ | 2 | 2 | 2 | 2 |
| | max depth | 20 | 20 | 20 | 20 |
| | unit time | 1s | 1s | 2s | 5s |

The encoder of the VQ-VAE is implemented as a two-layer GRU with a hidden dimension of 64, which processes the opponent trajectories and maps them into continuous embeddings $z \in \mathbb{R}^{32}$. Each embedding is quantized via nearest-neighbor assignment to a codebook with $K{=}16$ entries, producing a discrete latent type $e_k$. The decoder is a GRU-based autoregressive model conditioned on $e_k$; its hidden state is initialized from a learned projection of $e_k$, and it outputs step-wise likelihoods of opponent actions $\tilde{\pi}_k(a_{-i} \mid h)$. The quantizer is trained using the standard VQ-VAE objective, which combines reconstruction, codebook, and commitment losses, optimized with a straight-through estimator and exponential-moving-average codebook updates.

All experiments were executed on two dedicated workstations. For the PE, PP, and RWS environments we used an Intel Core i7-12700KF (12th Gen, 3.60 GHz) paired with an NVIDIA GeForce RTX 4060 Ti. The One-on-One experiments ran on a server equipped with an Intel Xeon E5-2620 v4 (2.10 GHz, 8 cores) and an NVIDIA TITAN Xp.

# D   Extra Results

## D.1   Scalability

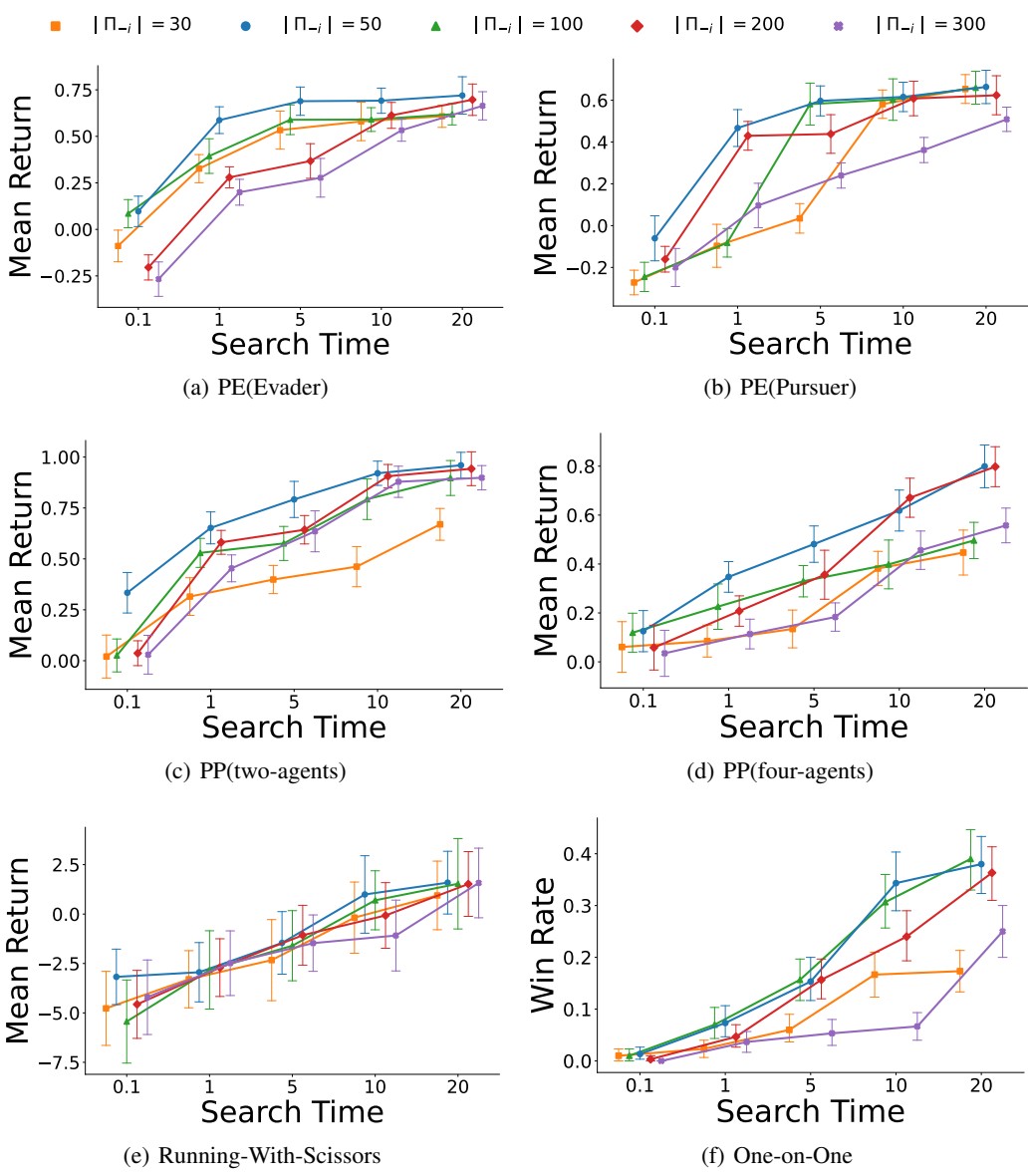

Figure 8: Scalability with policy library size

Figure 8 tracks how our planner behaves when we give it more thinking time while also varying how many opponent policies it must consider. When the library is tiny ($|\Pi_{-i}| = 30$), the curve levels off early: there simply are not enough opponent styles in the set, so the agent soon hits a ceiling. At the other extreme ($|\Pi_{-i}| = 300$), the first few points are lower because the agent must sift through many more hypotheses, but as the time budget grows this setting catches up and finally matches the mid-sized libraries. Libraries of $50 - 100$ policies combine wide coverage with a moderate inference cost, so they reach high returns the fastest across all six tasks. Most importantly, every curve keeps climbing and the error bars stay narrow, which means the belief update of QOM remains stable even when the number of possible opponents grows.

## D.2 Ablation of Belief Temperature $\beta$

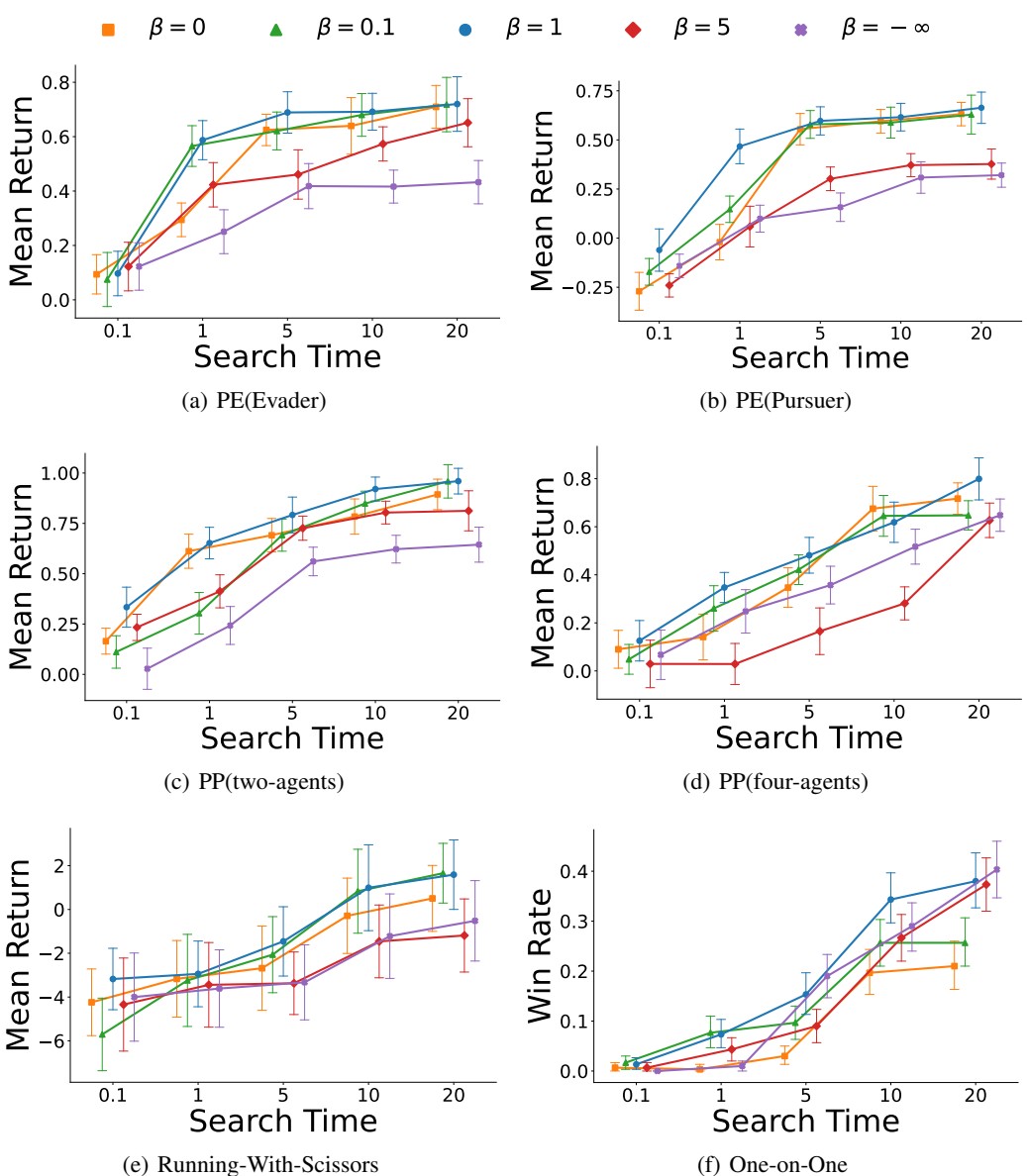

Figure 9: Ablation study on belief temperature $\beta$

For belief temperature $\beta$, we conduct additional ablation across all environments Figure 9. Across all tasks, moderate entropy levels (e.g., $\beta = 1$) consistently lead to stronger performance, confirming the generality of the trend observed in the main text. Both overly sharp posteriors ($\beta = -\infty$) and overly diffuse ones ($\beta = 0$) result in degraded returns or win rates. This highlights the importance of planning in diverse strategic settings.

## D.3 Ablation of Opponent Simulation Policy $\tilde{\pi}_{-i}^{(k)}$

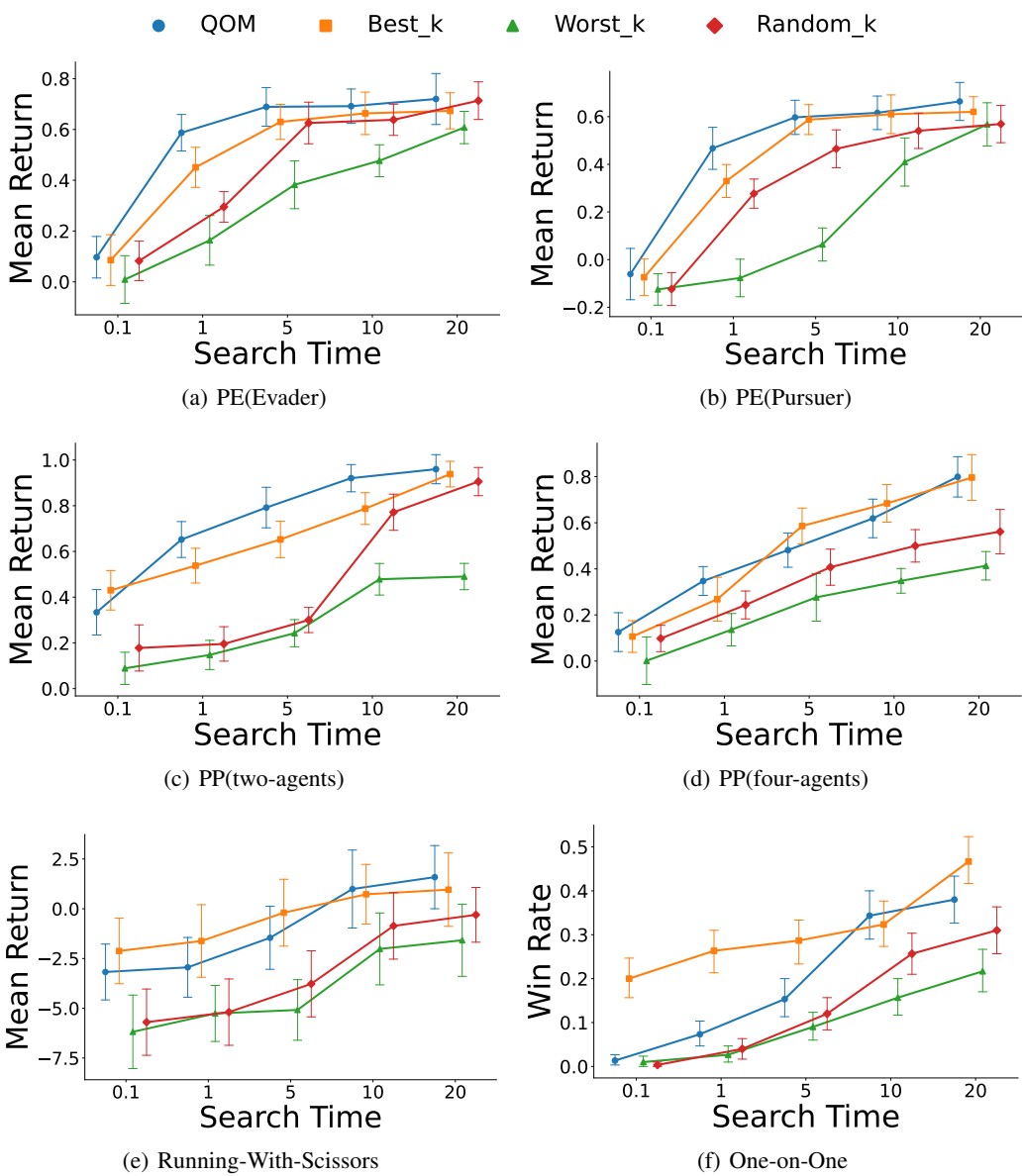

Figure 10: Ablation study on opponent simulation policy $\tilde{\pi}_{-i}^{(k)}$

Figure 10 shows how different ways of using the latent type information affect planning quality. QOM reaches higher scores more quickly than the other three choices and keeps that lead until the longest time horizon. The advantage is largest when the time budget is short: with only 0.1 or 1 unit of thinking time, QOM already gives a clear positive return, while Worst-k remains near zero and Random-k rises only slowly. Variability across runs is also lower for QOM, indicating that sampling from the belief offers a more stable balance between exploration and exploitation than any single-type shortcut.

## D.4 Ablation of Quantization Methods

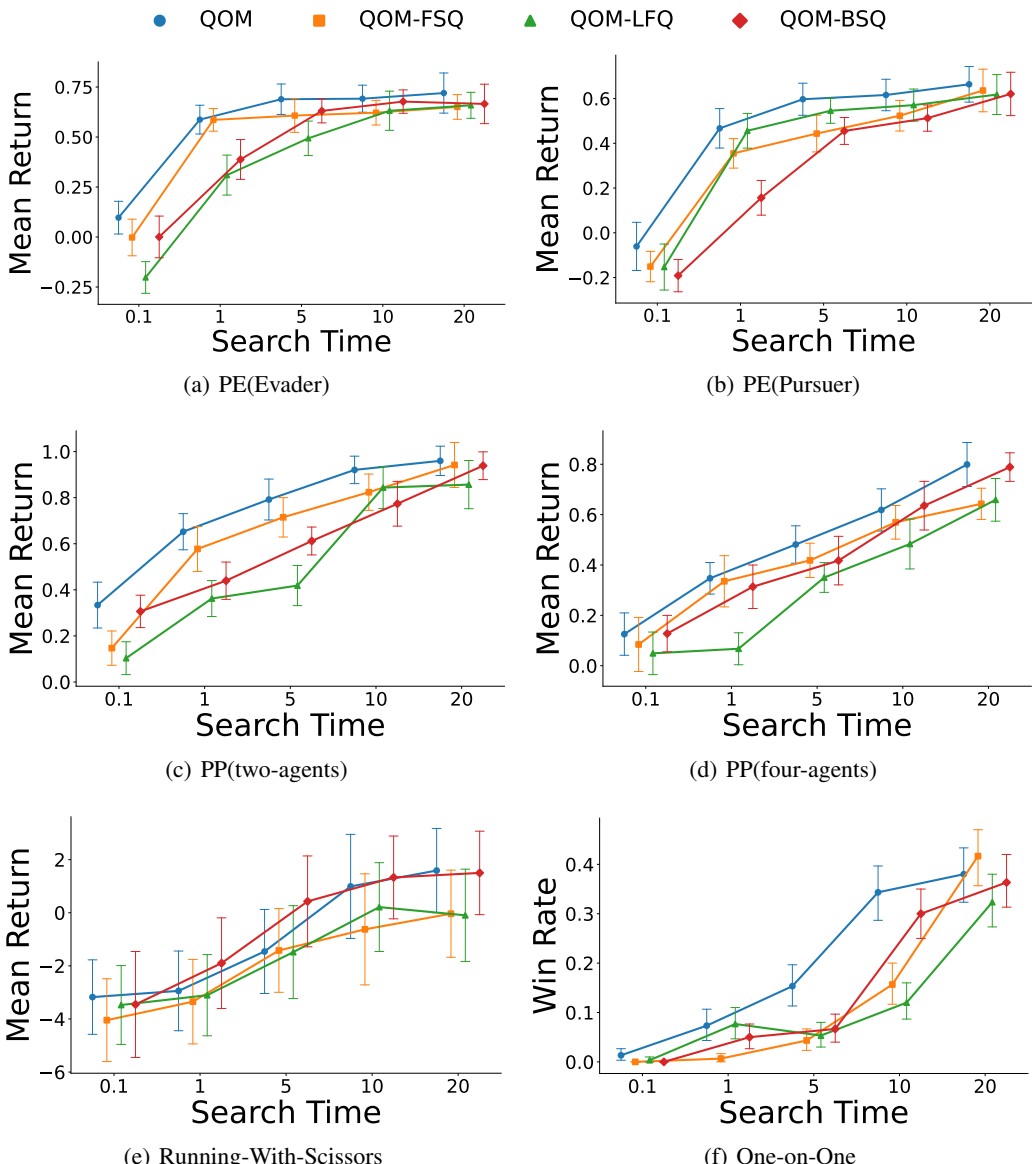

Figure 11: Ablation study on quantization methods

For different quantization methods we keep the planning code unchanged and only swap the vector-quantizer with its compressed variants FSQ, LFQ, and BSQ. Figure 11 shows that heavier compression costs performance. The gap is most visible in pursuit–evasion and the two-agent patrol task, where fine-grained cues about the opponent are important early on. As the search budget grows, all curves move closer, yet QOM still ends on top or tied by the last point. Overall, the experiment confirms that keeping more information in those codes yields higher and more stable rewards, while stronger compression trades some of that gain for smaller codebooks.

## D.5 Ablation of latent type size $K$

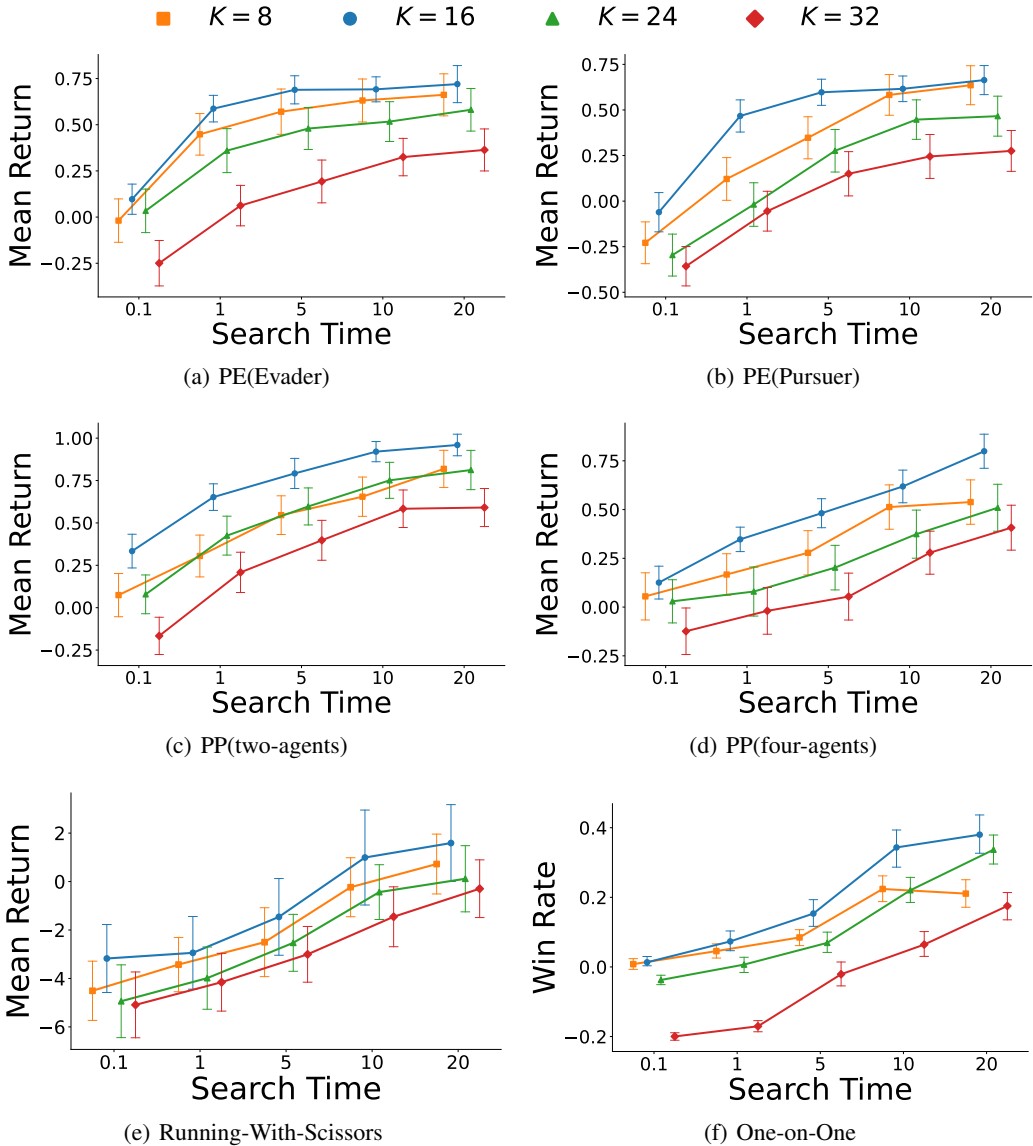

Figure 12: Ablation study on latent type size $K$

Figure 12 studies how the number $K$ of latent opponent types influences performance. When the codebook holds 16 types the agent posts the highest scores in every game and at every search-time setting. Using only 8 types comes close, but loses a few points once the search grows longer, suggesting that some opponent behaviors are not represented. When the codebook contains more latent types, early returns dip, and even with extra search time the larger setting remains behind the smaller ones. These results point to a simple trade-off: a small $K$ can miss important strategies, while a very large $K$ spreads the belief over too many possibilities, making each simulation less focused. A medium value offers enough variety to recognise the opponent without overloading the search, giving the best balance between coverage and efficiency.

