# OpenReview forum: "Planning with Quantized Opponent Models"
_NeurIPS.cc/2025/Conference — NeurIPS 2025 poster_

### Official Review · Reviewer_PHgn · 2025-07-03

**Clarity:** 2
**Significance:** 3
**Originality:** 2
**Rating:** 4
**Confidence:** 4

**Summary:**

This paper proposes a new framework called Quantized Opponent Models (QOM) for planning in multi-agent environments with unknown and potentially non-stationary opponents. The core idea is to discretize the opponent policy space using a quantized autoencoder (VQ-VAE), mapping each policy to a latent type. During interaction, the agent maintains a Bayesian belief over these types. It then constructs a belief-weighted soft best-response policy, known as the meta-policy, and uses it to guide action selection within a Monte Carlo Tree Search (MCTS) planner. This design allows opponent modeling to be integrated directly into the planning process.

The paper presents a theoretical result showing that the posterior over opponent types concentrates under bounded quantization error, and evaluates the method across several multi-agent environments, including adversarial and partially observable settings. Empirically, QOM consistently outperforms baseline methods, especially under limited computational budgets, and maintains robust performance when facing switching opponents or previously unseen policies.

**Questions:**

1. How stable is the belief distribution $b_t(k)$ over short time horizons, and when the number of types $K$ is large and closely spaced, how reliably can the model distinguish between them?
2. During MCTS rollouts, how is the opponent’s observation $o_{-i}$ inferred or approximated, given that it is not directly observable to the agent?
3. In the “Adaptation to Switching Opponents” experiment, what exactly does “tag failure” refer to, and how does it determine when the opponent switches its policy?
4. How are the unseen opponent policies constructed in the generalization experiments, and if the training set already covers diverse PSRO checkpoints, what makes these test-time opponents meaningfully novel?
5. The ablation on quantization methods compares several discrete encoding schemes (e.g.,FSQ, LFQ, BSQ), but since the performance differences are small, what insight is this experiment meant to provide regarding the role of quantization in the overall method?
6. Why is evaluation almost entirely based on performance versus search time, rather than more direct metrics like wall-clock runtime or tree size that would reflect computational cost?

**Ethical Concerns:**

["NO or VERY MINOR ethics concerns only"]

**Final Justification:**

The authors' response addressed some of my concerns and confusion regarding the methodological details. However, the use of MCTS in the planning module weakens QOM's applicability in partially observable environments. Therefore, I have decided to keep my original score unchanged.

**Limitations:**

The authors acknowledge some key limitations, such as the reliance on a fixed opponent strategy library and the assumption of piecewise-stationary behavior during interaction.

**Quality:**

3

**Strengths And Weaknesses:**

**Strengths**
1. The proposed method of discretizing the opponent policy space using a quantized autoencoder is conceptually simple, and avoids the need for handcrafted type definitions.
2. The integration of opponent modeling into the planning loop—through a belief-weighted prior in MCTS—is well-motivated and cleanly implemented.
3. The empirical evaluation spans a reasonably diverse set of environments and includes ablation studies that clarify the impact of key design components.

** Weaknesses**
1. I find your framework is quite similar to [1]. Like your work, [1] maintains beliefs about agents' types, includes an opponent simulation component (goal-conditioned policy in [1]) to predict opponents’ actions, and features a planning module to determine the best response. Both approaches use Bayes’ rule to update beliefs, with the opponent simulation component also acting as the likelihood term in Bayes' rule. Additionally, the opponent simulation component is used to simulate others’ actions during planning. Unlike [1], you use a pre-trained VQ-VAE to infer the types of opponents, while [1] represents opponent types using goals selected from a fixed goal set. Consequently, your approach requires both an agent library and an opponent library, as well as many samples to pre-train the VQ-VAE. In contrast, [1] requires expert knowledge to define a goal set for each environment. Additionally, you employ MCP for planning, while [1] uses MCTS. However, I may have overlooked some differences. Please clarify the novelty and advantages of your approach compared to [1], and ensure you cite it appropriately.
2. The approach relies heavily on a fixed offline opponent strategy library, which may limit its applicability in settings where the opponent pool is open-ended or evolving.
3. There is limited discussion of the computational cost introduced by belief updates, latent inference, and payoff matrix construction. While QOM is claimed to be more efficient than baselines, no runtime comparisons or profiling are provided.
4. The method is evaluated only with pre-defined agents and policies. It is unclear how it would perform in more dynamic, co-adaptive environments where both agents and opponents are learning simultaneously.

[1] Huang, Yizhe, et al. "Efficient Adaptation in Mixed-Motive Environments via Hierarchical Opponent Modeling and Planning." International Conference on Machine Learning. PMLR, 2024.

---

> ### Author Rebuttal · Authors · 2025-07-30
>
> Thank you for acknowledging our contributions as well as raising valuable questions.
>
> > I find your framework is quite similar to [1]. Like your work, [1] maintains beliefs about agents' types, includes an opponent simulation component (goal-conditioned policy in [1]) to predict opponents’ actions, and features a planning module to determine the best response. Both approaches use Bayes’ rule to update beliefs, with the opponent simulation component also acting as the likelihood term in Bayes' rule. Additionally, the opponent simulation component is used to simulate others’ actions during planning. Unlike [1], you use a pre-trained VQ-VAE to infer the types of opponents, while [1] represents opponent types using goals selected from a fixed goal set. Consequently, your approach requires both an agent library and an opponent library, as well as many samples to pre-train the VQ-VAE. In contrast, [1] requires expert knowledge to define a goal set for each environment. Additionally, you employ MCP for planning, while [1] uses MCTS. However, I may have overlooked some differences. Please clarify the novelty and advantages of your approach compared to [1], and ensure you cite it appropriately. [1] Huang, Yizhe, et al. "Efficient Adaptation in Mixed-Motive Environments via Hierarchical Opponent Modeling and Planning." International Conference on Machine Learning. PMLR, 2024.
>
> While both QOM and HOP [1] involve opponent modeling and planning, the two methods differ fundamentally in their problem formulation and modeling assumptions. HOP relies on a structured environment with an explicit goal space, modeling opponent behavior via goal inference and goal-conditioned policies. This hierarchical formulation is central to its mechanism and restricts its applicability to settings where goals are well-defined and observable. In contrast, QOM assumes no access to such goal structures. Instead, it operates in general partially observable multi-agent environments, modeling opponents through latent policy types learned via a quantized autoencoder. The agent maintains a Bayesian belief over these types, enabling efficient and scalable planning without relying on predefined goals. Thus, QOM addresses a more general setting and offers a flexible solution applicable across a wider range of domains where goals may be implicit or unavailable.
>
> > The approach relies heavily on a fixed offline opponent strategy library, which may limit its applicability in settings where the opponent pool is open-ended or evolving.
>
> We would like to clarify that this reliance on an offline opponent policy library is a common feature shared by many existing approaches, and such methods can still be effective in practice—provided that the strategy library sufficiently covers the policy space. This highlights an important but often overlooked aspect: the quality and diversity of the policy library are critical. Our proposed method, QOM, partially addresses this issue by reducing the reliance on handcrafted opponent strategies. Instead of manually designing diverse policies, QOM leverages quantitative objective modeling to construct a structured opponent representation space, from which diverse and representative policies can be more effectively identified and utilized.
>
> Regarding settings with open-ended or evolving opponents, we acknowledge that this remains an open challenge for the community. While our current framework assumes a fixed opponent pool, it can be extended in future work to incorporate adaptive mechanisms such as online strategy expansion or continual opponent modeling. We leave it for future work.
>
> > There is limited discussion of the computational cost introduced by belief updates, latent inference, and payoff matrix construction. While QOM is claimed to be more efficient than baselines, no runtime comparisons or profiling are provided.
>
> The payoff matrix construction is performed offline, which removes a substantial computational burden from the online decision-making loop. While belief updates and latent inference are executed online, they are designed to be lightweight and incremental, enabling fast and efficient planning during interaction. Our efficiency claim focuses on the online planning process, specifically in terms of search budget utilization. Compared to baselines, QOM achieves higher performance given the same per-step search budget. We will make this distinction explicit in the revision.
>
> > The method is evaluated only with pre-defined agents and policies. It is unclear how it would perform in more dynamic, co-adaptive environments where both agents and opponents are learning simultaneously.
>
> While most of our experiments are based on a fixed policy library, we would like to point out that we do consider partially dynamic settings—for example, in Figure 3(a), the opponent switches its policy during interaction, requiring the agent to adapt. We agree that fully co-adaptive environments, where both agents and opponents learn simultaneously, are more challenging. This remains a limitation for library-based approaches such as QOM and POTMMCP. However, when the policy library is sufficiently diverse and well-structured, it can provide strong coverage of the policy space. We believe this enables the agent to handle a wide range of dynamic opponent behaviors.
>
> > How stable is the belief distribution $b_t(k)$ over short time horizons, and when the number of types $K$ is large and closely spaced, how reliably can the model distinguish between them?
>
> From our empirical observations, when using a reasonably chosen number of types $K$, the belief distribution $b_t(k)$ typically shows preference for several likely types after a few observations, enabling effective and focused planning. We will add a visualization of belief dynamics over time horizons in the revision. Conversely, setting $K$ too large tends to over-fragment the latent space, spreading the belief across many similar types and thereby diluting the planning focus in each simulation. This phenomenon is discussed in Figure 11, where we show that excessively large $K$ leads to diminishing returns and even degraded performance due to type redundancy.
>
> > During MCTS rollouts, how is the opponent’s observation $o_{-i}$ inferred or approximated, given that it is not directly observable to the agent?
>
> The opponent’s observation $o_{-i}$ is approximated during rollouts using the environment simulator, which generates plausible observations from simulated joint actions and next states. In each rollout step, we use the environment simulator to generate $o_{-i}$ by stepping forward with the sampled joint action, and update the opponent’s history accordingly. This iterative process enables belief tracking and simulation without directly observing private opponent observations.
>
> > In the “Adaptation to Switching Opponents” experiment, what exactly does “tag failure” refer to, and how does it determine when the opponent switches its policy?
>
> In the Running-with-Scissors environment, “tag” is an action where the opponent marks a 3×3 zone ahead to challenge the agent, with outcomes decided by RPS rules. If the game doesn't end, the opponent switches policy. We will clarify this in the revision.
>
> > How are the unseen opponent policies constructed in the generalization experiments, and if the training set already covers diverse PSRO checkpoints, what makes these test-time opponents meaningfully novel?
>
> The unseen opponent policies used in the generalization experiments are generated from three additional rounds of PSRO training, independent of those used to construct the QOM training policy set. These additional policies are not included in the quantized opponent model, and provide novel strategic behaviors that differ from the training set due to differences in PSRO dynamics across runs. This setup allows us to assess generalization beyond the fixed policy catalog. We will clarify this detail in the revision.
>
> > The ablation on quantization methods compares several discrete encoding schemes (e.g.,FSQ, LFQ, BSQ), but since the performance differences are small, what insight is this experiment meant to provide regarding the role of quantization in the overall method?
>
> The experimental results indicate that QOM achieves consistently performance across different quantization schemes, with only marginal differences. This suggests that the effectiveness of QOM does not heavily depend on the specific choice of quantization strategy, highlighting its robustness and modularity — a valuable property for practical deployment.
>
> > Why is evaluation almost entirely based on performance versus search time, rather than more direct metrics like wall-clock runtime or tree size that would reflect computational cost?
>
> We would like to clarify that our evaluation is based on wall-clock time. As stated in Section 4.2, the “search time budget” refers to real wall-clock runtime limits per decision step (e.g., 0.1s, 1s, 5s, etc.), which are implemented as actual timing constraints during online planning. To make this more concrete, the Hyperparameter table in Appendix C lists the precise wall-clock durations (in seconds) used as unit time for each environment. These are empirically calibrated to reflect the relative complexity of each domain and ensure comparability across methods. POTMMCP is evaluated under the same wall-clock constraints, ensuring a fair and consistent comparison. We will clarify this more explicitly in the revision.

---

> > ### Comment · Reviewer_PHgn · 2025-08-02
> >
> > Thanks for the authors' detailed response, which addresses most of my confusion and concerns.
> >
> > With regard to the comparison between QOM and HOP, I completely agree that QOM addresses a more general setting by leveraging VQ-VAE to infer latent agent types. This distinction is something I already highlighted in my previous review. Since you have not emphasized any additional novelty or advantages of your approach, I maintain that the other components of your method (belief tracking, opponent policy modeling, and action planning)  have corresponding modules with HOP and employ similar techniques (e.g., Bayesian updates, Monte Carlo sampling under opponent uncertainty with pUCT) as those in HOP. Therefore, I strongly encourage the authors to provide a thorough discussion of HOP and this line of works in the Related Work section, while also elaborating on your generality advantage, which I agree is an important advantage.
> >
> > Regarding the MCTS rollouts, let me clarify my confusion: The simulator generates $(s', o, r) \sim G(s, a_i, a_{-i})$, but how do you set the simulator to the true state $s$ at the start of planning, given that $o_i^{(t)} \neq s$ due to partial observability? Are you assuming that the simulator always has access to the true state $s$?

---

> > > ### Author Response · Authors · 2025-08-03
> > > **Response to Reviewer PHgn**
> > >
> > > We sincerely appreciate the reviewer's follow-up response.
> > >
> > > > With regard to the comparison between QOM and HOP, I completely agree that QOM addresses a more general setting by leveraging VQ-VAE to infer latent agent types. This distinction is something I already highlighted in my previous review. Since you have not emphasized any additional novelty or advantages of your approach, I maintain that the other components of your method (belief tracking, opponent policy modeling, and action planning) have corresponding modules with HOP and employ similar techniques (e.g., Bayesian updates, Monte Carlo sampling under opponent uncertainty with pUCT) as those in HOP. Therefore, I strongly encourage the authors to provide a thorough discussion of HOP and this line of works in the Related Work section, while also elaborating on your generality advantage, which I agree is an important advantage.
> > >
> > > Thank you for the helpful feedback. We will include a concise discussion of HOP and related works, and clarify the generality advantage of our method.
> > >
> > > > Regarding the MCTS rollouts, let me clarify my confusion: The simulator generates $(s', o, r) \sim G(s, a_i, a_{-i})$, but how do you set the simulator to the true state $s$ at the start of planning, given that $o_i^{(t)} \ne s$ due to partial observability? Are you assuming that the simulator always has access to the true state $s$?
> > >
> > > Thank you for raising this important point. In our ground-truth environments (Pursuit–Evasion, Predator–Prey, Running-with-Scissors), we assume access to the latent initial state $s_0$ at each real decision step and deterministic transition dynamics. Under this setting, the external simulator used for planning is reset to the current latent state $s$, and MCTS rollouts proceed by deterministically stepping the simulator with sampled joint actions to produce the next state and observations, i.e., $(s', o_i, o_{-i}, r) = G(s, a_i, a_{-i})$. Because dynamics are deterministic, knowing $s_0$ and the realized joint actions would suffice to reproduce the exact state trajectory in replay. For the One-on-One environment, we initialize rollouts from a fixed state of the learned dynamics model and generate opponent observations through that model, matching prior work. We acknowledge that such setting weakens the practical effect of partial observability during planning. We will revise the paper to state this explicitly, clarify which environments use ground-truth versus learned simulators, and discuss the implications and limitations, along with how one could handle stricter partial observability by initializing search from a belief over states rather than the true state.

---

### Official Review · Reviewer_PqEB · 2025-07-03

**Clarity:** 3
**Significance:** 3
**Originality:** 3
**Rating:** 5
**Confidence:** 3

**Summary:**

The paper addresses a core challenge for partially observed multi-agent environment, that is the uncertainty about an opponent's behavior. It proposes a framework where a "catalog" of opponent "strategies" is learned from data by using VQVAE. This allows an agent to maintain a Bayesian belief over a finite set of opponent "strategies" and update these beliefs at test time. During test time, the framework leverages the opponent model for planning in MCTS. Experimental results show strong performance compared to other baselines.

**Questions:**

1. Could the author clarify whether a ground-truth or learned simulator is used in the experiment?

2. Bayesian update is used for updating belief over opponents. Have the author experiment with other belief updates such as expert switching [1], which alleviates the catch-up phenomenon in Bayesian update [2]?

3. The paper mentions using a VQ-VAE and other alternatives to learn opponent "strategies". Could you elaborate on the architecture of the quantizer?

4. How well is the learned "strategies" mapped to the opponent's policy library? And does a "good" mapping contributes to better performance? Could you provide some discussion?

5. Since the framework is quite complex, will the author open source the code?

[1] W. M. Koolen et al. Combining expert advice efficiently.

[2] T. van Erven et al. Catching Up Faster in Bayesian Model Selection and Model Averaging.

**Ethical Concerns:**

["NO or VERY MINOR ethics concerns only"]

**Final Justification:**

The rebuttal has addressed my concern. I maintain the score, but downgrade the confidence level as I feel other reviewers are more familiar with the multi-agent frameworks than me.

**Limitations:**

Yes. More societal impact could be discussed.

**Quality:**

4

**Strengths And Weaknesses:**

**Strength**

Overall the paper is well-written. It provides a good description for a very complex framework. Theoretical aspect is sound.

Experimental results are strong. The proposed Quantized Opponent Models (QOM) consistently outperform state-of-the-art baselines across a variety of multi-agent environments. Ablations are extensive.

**Weaknesses**

The proposed framework requires extensive offline preparation and potentially quite expensive in terms of compute.

The number clusters learned by VQVAE is still quite small, therefore, it's questionable whether the framework can be extended to work in more complex environments.

---

> ### Author Rebuttal · Authors · 2025-07-30
>
> Thank you sincerely for your feedback and appreciation.
>
> > Could the author clarify whether a ground-truth or learned simulator is used in the experiment?
>
> We use ground-truth simulators for the Pursuit-Evasion, Predator-Prey, and Running-with-Scissors environments. For the One-on-One environment, we follow the common practice in prior work (MBOM) and use a learned simulator. This setup ensures consistency with the baseline implementations and allows for fair performance comparison. We will clarify this distinction in the revision.
>
> > Bayesian update is used for updating belief over opponents. Have the author experiment with other belief updates such as expert switching [1], which alleviates the catch-up phenomenon in Bayesian update [2]? [1] W. M. Koolen et al. Combining expert advice efficiently. [2] T. van Erven et al. Catching Up Faster in Bayesian Model Selection and Model Averaging.
>
> While we do not use expert‑switching updates, we explicitly address the catch‑up phenomenon with a lightweight forgetting mechanism. After each Bayes step (Eq. (5)) we apply belief smoothing by interpolating the posterior with a uniform prior, $b^{\text{smooth}}_{t+1}=(1-\lambda)\,b_{t+1}+\lambda\,b_0$. This constant‑share (“leak”) step (i) maintains a non‑vanishing probability mass on alternative types, preventing premature lock‑in, and (ii) induces an effective memory of roughly $1/\lambda$ steps, which shortens recovery after occasional switches or model mismatch. Its effect is largely neutral in very short interactions, but in piecewise‑stationary scenarios it reduces catch‑up time without changing the rest of the planner. Expert‑switching priors are a valuable direction and may further help in our Adaptation to Switching Opponents setting (Sec. 4.4). We will include a concise comparative experiment in the revision.
>
> > The paper mentions using a VQ-VAE and other alternatives to learn opponent "strategies". Could you elaborate on the architecture of the quantizer?
>
> The encoder of VQ‑VAE is a 2‑layer GRU sequence model (hidden size 64) that reads the opponent trajectory and maps it to a continuous embedding $z\in\mathbb{R}^{32}$. We vector‑quantize $z$ by nearest‑neighbor assignment to a $K$-entry codebook ($K{=}16$), yielding a discrete type $e_k$. The decoder is a GRU‑based autoregressive model conditioned on $e_k$: we initialize the decoder hidden state with a learned projection of $e_k$ and predict per‑step opponent‑action likelihoods $\tilde\pi_k(a_{-i}\!\mid h)$. The quantizer is trained with the standard objective (reconstruction + codebook + commitment losses) using a straight‑through estimator with EMA codebook updates. We will revise and expand the appendix to include these architectural details and hyperparameters.
>
> > How well is the learned "strategies" mapped to the opponent's policy library? And does a "good" mapping contributes to better performance? Could you provide some discussion?
>
> As shown in Figure 6, we visualize the payoff matrices between (i) agent policies and latent types (the learned codebook), and (ii) agent policies and opponent policies. However, as you pointed out, these matrices alone do not make the policy–type correspondence explicit. To address this, we directly measure behavioral similarity between opponent policies and latent types. While figures cannot be shown here, the analysis shows that more than half of the opponent policies align strongly with a single latent type, whereas the remainder spread their alignment over two to three types (multimodal behavior). We will include the corresponding illustration and analysis in the revision.
>
> > Since the framework is quite complex, will the author open source the code?
>
> Yes, we plan to release the code, and the link will be available in the revision.

---

### Official Review · Reviewer_VDJ5 · 2025-07-03

**Clarity:** 3
**Significance:** 3
**Originality:** 3
**Rating:** 4
**Confidence:** 4

**Summary:**

This paper presents a method for multi-agent planning by learning a finite set of opponent types from trajectory data using a VQ-VAE. The learning of this quantized autoencoder bases on a simulated offline dataset trajectories of the agents' interactions with the knowledge of the behavior policy libraries of both the agent and the opponents. The agent updates their belief over the opponents' type after this offline training, and makes decision with expectation on sampled types from this belief together with per-type policies derived from a pre-trained payoff matrix. Experiments on some multi-agent environments show that the proposed method achieves an effective performance with less searching cost.

**Questions:**

Please see the weakness section above.

**Ethical Concerns:**

["NO or VERY MINOR ethics concerns only"]

**Final Justification:**

I am keeping my rating and would like the authors to provide some additional experiments in the revision.

**Limitations:**

Yes

**Quality:**

3

**Strengths And Weaknesses:**

## Strengths

* The idea of modeling opponents' behavior into finite patterns via a vector-quantized autoencoder is interesting and novel.

* The proposed method has all the merits inherited from type-based methods, Beyesian reasoning, and Monte-Carlo planning, while avoiding the drawbacks of manually defined behavior classes and sample inefficiency during online planning.

* Ablation study shows reasonable and interpretable explanations over different design choices.

## Weaknesses

* Major weaknesses

  * The success of the method depends heavily on (1) the quality of the codebook of the opponents' behavior from the quantized encoder trained with the offline trajectories; (2) the payoff matrix estimated from the offline trajectories. Any illustrations of the learned codebook and their correspondence with the opponent policies would be helpful to understand the quantization process.

  * The success of the proposed method also relies on a good set of opponent policies that are close to the true opponents' behavior in real-time execution. However, the paper does not provide enough demonstration showing that with the proposed method's planning, the agent also faces similar opponents' behavior as in the offline training. It is claimed in this paper (at Line 231) that the opponent policy library has 50 policies including final trained opponents' policy and intermediate checkpoints. If I am understanding correctly, the opponent's policy can also include any agent's own individual policy and freely combine them as long as they share same state and action space. Therefore, the complexity of the opponent's policy library can be very high, depending exponentially on the number of agents.


* Minor weaknesses

  * In Figure 2 and 3, the x-axis does not represent the actual quantitative values along its direction. Each point is a different choice of search time but they are equally segmented. They are neither linear scale nor log scale. Therefore, the current plot does not reflect the actual true trends with respect to the increase of the search time. It would be better to use other representations than solid lines.

  * It is unclear to me for different search time, what is the actual wall time used for the search. Are the actual wall time requires for the search linear with respect to the search time? If not, it would be better to clarify this in the paper.

---

> ### Author Rebuttal · Authors · 2025-07-30
>
> Thank you for acknowledging our novel contributions as well as raising valuable questions. As follows, we address your concerns in detail.
>
> > The success of the method depends heavily on (1) the quality of the codebook of the opponents' behavior from the quantized encoder trained with the offline trajectories; (2) the payoff matrix estimated from the offline trajectories. Any illustrations of the learned codebook and their correspondence with the opponent policies would be helpful to understand the quantization process.
>
> As shown in Figure 6, we visualize the payoff matrices between (i) agent policies and latent types (the learned codebook), and (ii) agent policies and opponent policies. However, as you pointed out, these matrices alone do not make the policy–type correspondence explicit. To address this, we directly measure behavioral similarity between opponent policies and latent types. While figures cannot be shown here, the analysis shows that more than half of the opponent policies align strongly with a single latent type, whereas the remainder spread their alignment over two to three types (multimodal behavior). We will include the corresponding illustration and analysis in the revision.
>
> > The success of the proposed method also relies on a good set of opponent policies that are close to the true opponents' behavior in real-time execution. However, the paper does not provide enough demonstration showing that with the proposed method's planning, the agent also faces similar opponents' behavior as in the offline training. It is claimed in this paper (at Line 231) that the opponent policy library has 50 policies including final trained opponents' policy and intermediate checkpoints. If I am understanding correctly, the opponent's policy can also include any agent's own individual policy and freely combine them as long as they share same state and action space. Therefore, the complexity of the opponent's policy library can be very high, depending exponentially on the number of agents.
>
> We agree that many policy-library–based methods struggle to generalize to opponents outside the training population [1,2]. Classic BPR-style reuse relies on exact opponent matches and performs poorly when facing novel strategies. In contrast, QOM quantizes the opponent policy space into $K$ latent types and maintains a Bayesian posterior over them. This allows QOM to assign probability mass to nearby types and adaptively plan, even when the online opponent is not in the library. This generalization is demonstrated in our out-of-library test (Figure 3(a)).
>
> In term of complexity, we treat all opponents as a single composite agent with a joint action space; consequently, online planning reasons only over K latent opponent types, rather than over combinations of per‑opponent policies. Constructing separate policy libraries for each opponent and composing them at decision time would, as you point out, cause an exponential increase in complexity. Exploring such factorized libraries is outside our current scope and left for future work.
>
> [1] Jing Y, Liu B, Li K, et al. Opponent modeling with in-context search. NeurIPS, 2024.
>
> [2] Lian J, Huang Y, Ma C, et al. Fusion-PSRO: Nash policy fusion for policy space response oracles. arXiv:2405.21027.
>
> > In Figure 2 and 3, the x-axis does not represent the actual quantitative values along its direction. Each point is a different choice of search time but they are equally segmented. They are neither linear scale nor log scale. Therefore, the current plot does not reflect the actual true trends with respect to the increase of the search time. It would be better to use other representations than solid lines.
>
> Our plots could misleadingly suggest a continuous quantitative x‑axis. In the revision, we will re-plot the figures by treating search time as a categorical variable and will remove all connecting solid lines.
>
> > It is unclear to me for different search time, what is the actual wall time used for the search. Are the actual wall time requires for the search linear with respect to the search time? If not, it would be better to clarify this in the paper.
>
> We would like to clarify that our evaluation is based on wall-clock time. As stated in Section 4.2, the “search time budget” refers to real wall-clock runtime limits per decision step (e.g., 0.1s, 1s, 5s, etc.), which are implemented as actual timing constraints during online planning. The precise durations used for each environment are listed in the Hyperparameter Table in Appendix C. In practice, the actual wall time used scales approximately linearly with the search time budget. POTMMCP is evaluated under the same wall-clock constraints, ensuring a fair and consistent comparison. Thank you for the suggestion and we will clarify this more explicitly in the revision.

---

> > ### Comment · Reviewer_VDJ5 · 2025-08-08
> >
> > Thanks for the authors' detailed response. I would like to encourage the authors to provide some preliminary analysis regarding using separate policy libraries for each opponent and composing them at decision time as this is a practical and reasonable scenario. I am keeping my rating now.

---

### Official Review · Reviewer_ETPj · 2025-07-03

**Clarity:** 3
**Significance:** 2
**Originality:** 3
**Rating:** 3
**Confidence:** 4

**Summary:**

This paper proposes a method for planning in multi-agent stochastic games. The method learns an encoder that maps opponents' policies into a discrete set of latent vectors, each representing a distinct agent archetype. Importantly, this embedding space is significantly smaller than the original policy space. Together with the encoder, the method also trains a decoder that reconstructs a policy for each latent vector. The planner then uses the decoded policies to compute a belief over the discrete set of agent types, based on the likelihood of the opponents' action history. This belief is then used to sample policies and plan a best response. The empirical results reported in Figure 2 suggest that the proposed method outperforms the selected baselines.

**Questions:**

1. According to Algorithm 1, the opponent’s policy may change within a single rollout. Wouldn’t it make more sense to sample an opponent policy once at the root node and keep it fixed for the entire rollout? As it stands, the planner could be reasoning over trajectories that do not correspond to any specific opponent type, but rather to a mixture of types, which may not be meaningful or valid for planning.

2. The paragraph starting at Line 169 explains that, to mitigate the effects of noisy observations or approximation errors, a smoothing operation is applied after the belief update to keep the belief close to the initial belief. Could the authors clarify why noise or approximation error would affect the belief in this way? This seems counterintuitive, since the belief is computed exactly using Bayes’ rule, and while noise may influence the observations, it should not affect the likelihood of actions under the decoded policies. Additionally, the ablation study ("Effect of Belief Temperature") does not clearly demonstrate the utility of this smoothing, as the error bars in Figure 4a appear to overlap significantly.

3. The idea of constructing a meta-policy seems somewhat orthogonal to the main contribution of the paper. Could the authors clarify the motivation for including this component and how it connects to the core idea of quantized opponent models? Additionally, in Line 205, the authors argue that using the meta-policy should improve sample efficiency. While I agree that, if the set of agent policies is good, planning efficiency may improve, it seems that this merely shifts the computational burden from online search to offline policy training and payoff matrix estimation. Could the authors comment on this trade-off?

**Ethical Concerns:**

["NO or VERY MINOR ethics concerns only"]

**Final Justification:**

After reading the authors’ response, I still stand by my assessment. The initial experimental section was weak (e.g., overlapping error bars), lacked important baselines for comparison, and included strong claims that were not adequately supported by the results. The authors addressed some of these issues in their response and provided preliminary results. However, given that the changes required to the original submission would be quite substantial, I believe a new round of review would be necessary.

**Limitations:**

Limitations are covered in the last section.

**Quality:**

2

**Strengths And Weaknesses:**

**Strengths**
The paper tackles an important problem: modeling opponent policies to adapt and plan the best response to specific types of opponents. The idea of projecting opponent policies into a discrete set of policy archetypes is compelling and novel.

**Weaknesses**
* Experiments: This is, in my opinion, the paper's biggest weakness, as some of the claims in the experimental section are not fully supported by the empirical evidence. A few points:
  - There is significant overlap in the error bars in Figures 3 and 4, making it difficult to draw any definitive conclusions about the relative performance of the methods. Additional runs are needed to make such claims more reliable.
  - Given that the opponent library consists of only 50 policies, it would be worthwhile to include a baseline that uses the library policies directly to update the belief.
  - More details should be provided on how the agent and opponent policy libraries are constructed, along with evidence demonstrating that the policies are diverse.

* Technical concerns: Some of the algorithmic choices and problem formulation decisions are questionable. Specifically:
  - Sampling multiple opponent policies within a single trajectory rollout in Algorithm 1. See Question 1.
  - Belief smoothing as described in Line 169. See Question 2.
  - Meta-policy construction in Section 3.3 seems to shift the computation from offline to online. See Question 3.
  - The payoff matrix is defined for each combination of agent and opponent types, but not for each state of the environment. As a result, the meta-policy favors agent policies that perform well on average across states, which may not be optimal for the agent's current state.
  - Although the method targets partially observable stochastic games involving multiple agents, all other opponents are treated as a single composite agent, and only one belief is maintained across them. This raises the question of why the problem is not instead formulated as a two-agent game.

**Minor Comments**
- In the first ablation study, is $\beta$ intended to be $\lambda$?

**Related Work**
- A. Czechowski and F. A. Oliehoek, 2020. *Decentralized MCTS via Learned Teammate Models.*

---

> ### Author Rebuttal · Authors · 2025-07-30
>
> Thanks for your valuable comments. Below is a detailed response to your question, addressing each point individually. Owing to space constraints, we reference your comment using its opening words.
>
> > There is significant overlap ...
>
> The RWS environment is highly stochastic, resulting in large variance and wide confidence intervals. Even with 300 runs, QOM outperforms baselines in Figure 3 (Adaptation and Generalization). Increasing to 1000 runs tightens CIs and reinforces statistical robustness. Figure 4, which analyzes hyperparameter and component sensitivity, retains valid trends despite observed variance.
>
> We increased the number of runs to 1000. Results are shown below. This larger sample yields tighter intervals. Results for other environments will be added in the revision.
>
> **Table: Extended results for Figure 3(a) — Adaptation evaluation in RWS with 1000 runs**
> |Search Time|0.1|1|5|10|20|
> |---|---|---|---|---|---|
> |QOM|-3.91 ± 0.94|-2.82 ± 1.09|-1.53 ± 0.80|-0.09 ± 0.80|0.20 ± 1.07|
> |POTMMCP|-5.11 ± 1.08|-4.32 ± 0.79|-2.66 ± 0.89|-1.44 ± 1.01|-1.51 ± 0.90|
> |I-POMCP-PF|-5.60 ± 0.80|-4.34 ± 1.15|-3.85 ± 0.93|-3.54 ± 0.95|-2.93 ± 0.93|
> |MBOM|-4.08 ± 0.79|-3.15 ± 1.07|-1.44 ± 0.92|-1.03 ± 1.01|0.19 ± 0.95|
>
> > Given that the opponent library ...
>
> We added QOM-direct, which performs exact categorical Bayes updates over 50 policies without quantization, using the same PUCT prior, belief smoothing, and history reconstruction. Under low budgets, QOM significantly outperforms QOM-direct; under higher budgets, performance converges. This suggests QOM’s type compression enables faster belief updates and focused exploration, while QOM-direct suffers from expensive marginalization and slow posterior concentration when policies are not clearly separable.
>
> **Table: Performance comparison between QOM and QOM-direct in PE(Evader)**
> |Search Time|0.1|1|5|10|20|
> |---|---|---|---|---|---|
> |QOM|0.10 ± 0.08|0.59 ± 0.07|0.69 ± 0.08|0.69 ± 0.07|0.72 ± 0.10|
> |QOM-direct|-0.26 ± 0.09|0.01 ± 0.06|0.06 ± 0.10|0.12 ± 0.09|0.16 ± 0.09|
>
> > More details should be provided ...
>
> Policy libraries are built via PSRO: each round trains a best-response against a policy mixture. We obtain 10 agent and 10 opponent policies, all trained via PPO with shared architecture and hyperparameters. To broaden coverage, we add intermediate PPO checkpoints, resulting in 50 opponent policies spanning exploration to convergence. Details are in Appendix C.2.
>
> > According to Algorithm 1 ...
>
> We understand the concern that, under Algorithm 1, a single simulation rollout may involve opponent actions that are generated from different latent types, rather than a single consistent policy throughout. This is indeed a deliberate design choice in QOM.
>
> While it is common in type-based planning methods (e.g., POTMMCP) to assume that the opponent adheres to a single fixed strategy sampled from a predefined library, QOM adopts a fundamentally different modeling philosophy. Rather than assuming the opponent follows a specific type from the quantized library, QOM maintains a posterior belief $b_t(k)$ over types and uses this belief to simulate opponent behavior at each decision step (Algorithm 1, line 7). In other words, each rollout samples an opponent action from a posterior‑predictive mixture, where the type is drawn per step according to current uncertainty. This simulates behavior from a belief‑weighted model $p(a_{-i}\mid h)=\sum_k b_t(k)\tilde\pi_k(a_{-i}\mid h)$, and is consistent with our modeling approach in Section 3.4 and Equation (7).
>
> There are two reasons we adopt this per-step marginalization. First, the quantized types in our library are not assumed to correspond to ground-truth opponent identities. Especially early in an episode, the posterior is uncertain or multi-modal, and marginalizing across likely types allows the planner to hedge its expectations. Second, this design is empirically justified: as shown in Figure 4(b), QOM — which marginalizes over types during simulation — outperforms heuristics that fix a specific opponent type during rollouts (e.g., “Best‑k”), across simulation budgets. In response, we will add a root-sampling variant as an ablation in the revision to evaluate its effect.
>
> > $\beta$ intended to be $\lambda$ ?
> > The paragraph starting at Line 169 ...
>
> Thank you for pointing out the notation clash. In the first ablation, the parameter is the belief temperature used in the Bayes update of the type posterior. We now make this explicit by tempering Eq. (5) as $b_{t+1}(k)\ \propto\ b_t(k)\,\big[\tilde\pi_k(a_{-i}^{(t)}\mid h_{-i}^{(t)})\big]^{\beta},$ so that $\beta=0$ recovers the uniform prior, $\beta=1$ yields the standard Bayes update, and $\beta\!\to\!\infty$ collapses to a greedy posterior. To avoid overloading notation, we use $\beta$ only in Eq. (5) and no longer in Eq. (2); the smoothing weight $\lambda$ is a separate post‑update mixing parameter and is unrelated to $\beta$.
>
> Regarding above “close to the initial belief,” this was a misunderstanding arising from the earlier notation issue. The purpose of belief smoothing is not to bias the posterior toward the prior per se, but to ensure that all types retain non‑zero probability mass. Although the update follows Bayes’ rule, the likelihoods are approximate: (i) they come from a learned decoder $\tilde\pi_k(a\mid h)$ over quantized types, and (ii) the opponent’s history $h_{-i}$ used by the decoder is reconstructed via the simulator under partial observability. Model misspecification and reconstruction error can make early posteriors spuriously sharp after a few highly discriminative observations; once certain types are driven near zero, recovery is difficult with finite samples. Smoothing therefore acts as a robust‑Bayes regularizer that avoids zero‑probability assignments while the agent is still gathering information. Our theoretical analysis also shows that the quantization error introduces an $O(\xi)$ uncertainty floor, which further justifies a small amount of robustness in the early phase.
>
> > The idea of constructing a meta-policy ...
>
> While it may appear that the meta-policy component is orthogonal to the main contribution of quantized opponent modeling, we emphasize that it is a critical mechanism for realizing belief-aware planning within our QOM framework. Specifically, the meta-policy plays three key roles. First, it guides action selection during Monte Carlo rollouts. The meta-policy serves as a belief-weighted mixture of soft best responses, which directly informs the agent’s action choices during Monte Carlo rollouts. Rather than relying on uninformed or uniform rollout policies, it provides a principled prior that focuses the search on strategies likely to succeed under the current opponent belief, thereby improving planning efficiency and robustness. Second, it provides a principled mechanism for responding to opponent uncertainty. Beyond rollout heuristics, the meta-policy operationalizes a soft best-response to the posterior over latent opponent types. This allows the agent to act adaptively under uncertainty—balancing exploitation and exploration—without hard type commitment or brittle switching. In essence, it connects the Bayesian belief over opponent behavior to real-time decision-making. Third, it serves as the interface between offline learning and online planning. The meta-policy leverages the offline-estimated payoff matrix, computed from agent-opponent policy interactions, to synthesize strategic behavior during planning. It thus serves as the bridge between offline opponent modeling (via quantized types) and online planning under uncertainty, aligning the agent’s strategy space with its evolving beliefs. The design of meta-policy is inspired by the idea introduced in POTMMCP. Our work builds upon and extends this concept by integrating it into a learned quantized belief model.
>
> In summary, the meta-policy is integral to the deployment of QOM: it enables efficient belief-aware planning by translating opponent uncertainty into informed action selection during planning. Without this mechanism, the benefits of the learned opponent models would not materialize in online decision-making. While this design shifts some computational burden to the offline phase, we believe it is a worthwhile and effective trade-off that significantly reduces online planning latency and improves real-time decision quality.
>
> > The payoff matrix is defined ...
>
> The payoff matrix is computed offline across trajectories, not conditioned on state. The meta-policy then acts as a dynamic prior in PUCT, while rollouts incorporate current history and belief. This hybrid yields efficient adaptation. Extending to state-conditional matrices via value approximators is a promising direction, and we leave it for future work.
>
> > Although the method targets ...
>
> We model all opponents as a single composite agent with a joint type belief—a common abstraction [1,2] to keep planning tractable in multi-agent settings. While a two‑player reformulation is possible, it would mask the genuinely multi‑agent interaction structure (e.g., coordination and interference among multiple pursuers in Predator–Prey) and reduce transfer to more general multi‑agent settings. Moreover, maintaining separate beliefs for $n$ opponents yields an exponential type space $K^n$, which is computationally prohibitive; a belief over joint types keeps complexity at $K$ while still capturing group behavior uncertainty. Our formulation therefore strikes a practical balance between expressiveness and tractability.
>
> [1] Anirudh Kakarlapudi, Gayathri Anil, Adam Eck, Prashant Doshi, and Leen-Kiat Soh. Decision theoretic planning with communication in open multiagent systems. PMLR, 2022.
>
> [2] Alessandro Panella and Piotr Gmytrasiewicz. Interactive pomdps with finite-state models of other agents. AAMAS, 2017
>
> > Related Work
>
> Thank you for the suggestion. We will include this work in the revised version.

---

> ### Comment · Reviewer_ETPj · 2025-08-02
>
> I would like to thank the authors for addressing some of my concerns. However, several issues still remain:
>
> **Overlapping Error Bars**
>
> I appreciate that the authors increased the number of runs to reduce the error bars. However, even with 1000 runs, there is a significant overlap between QOM and the MBOM baseline. This makes it difficult to support any claims that QOM significantly outperforms this baseline.
>
> **Computing the Belief Over the Policy Library**
>
> Thank you for running the additional experiments.
>
> > Under low budgets, QOM significantly outperforms QOM-direct; under higher budgets, performance converges.
>
> It seems that QOM still outperforms QOM-direct even at higher budgets (e.g., 10 and 20). Why is that the case? Since QOM-direct is exact, I would expect it to perform at least as well as QOM with a high budget.
>
>
> **Opponent Policy Sampling Strategy**
>
> > Especially early in an episode, the posterior is uncertain or multi-modal, and marginalizing across likely types allows the planner to hedge its expectations.
>
> It is unclear why sampling in every state is necessary to achieve this. If multiple rollouts are run, sampling at the root node still marginalizes over different opponent types.
>
> > Second, this design is empirically justified: as shown in Figure 4(b), QOM — which marginalizes over types during simulation — outperforms heuristics that fix a specific opponent type during rollouts (e.g., “Best-k”), across simulation budgets.
>
> In Figure 4(b), Best-k performance is very similar to QOM. Given the amount of overlap between the curves, it is not possible to draw any conclusions from this plot.
>
> **Meta-Policy**
>
> I remain unconvinced that the meta-policy is strictly necessary for QOM to function effectively. In the rebuttal, the authors list three reasons for its inclusion:
>
> 1. It guides action selection during Monte Carlo rollouts by focusing on strategies likely to succeed.
> 2. It is effective under opponent uncertainty by providing a soft best-response to possible opponents.
> 3. Leverages offline learning.
>
> I see points (1) and (2) as essentially the same, both aim to reduce variance in rollouts, while (3) is not a justification, but rather a description of the approach.
>
> More importantly, none of these reasons directly relate to the central idea of quantized opponent models. Including the meta-policy confounds the analysis, making it difficult to assess how much of the performance gain is due to the quantized models themselves versus the meta-policy.

---

> > ### Author Response · Authors · 2025-08-03
> > **Response to Reviewer ETPj （Part Ⅰ）**
> >
> > We sincerely thank you for the constructive feedback.
> >
> > > I appreciate that the authors increased the number of runs to reduce the error bars. However, even with 1000 runs, there is a significant overlap between QOM and the MBOM baseline. This makes it difficult to support any claims that QOM significantly outperforms this baseline.
> >
> > We will revise the wording in the paper to avoid overstating performance differences. Specifically, we will remove claims of "significant" improvement where the confidence intervals overlap in the revision.
> >
> > > It seems that QOM still outperforms QOM-direct even at higher budgets (e.g., 10 and 20). Why is that the case? Since QOM-direct is exact, I would expect it to perform at least as well as QOM with a high budget.
> >
> > In addition to the computational overhead and slower posterior concentration previously discussed, we believe the remaining performance gap, even under higher budgets, can be attributed to differences in inductive bias and inference dynamics.
> >
> > Specifically, QOM leverages VQ-VAE to learn a compact latent representation that clusters similar opponent policies. This quantization introduces a structural prior that can smooth over small behavioral variations across policies, effectively regularizing belief updates. Even though QOM is not performing exact inference over the full library, the compressed latent space can reduce ambiguity during planning by focusing on meaningful behavioral distinctions.
> >
> > In contrast, QOM-direct operates over all 50 policies directly, and although it performs exact Bayesian updates, **it may suffer from inference noise or slower belief sharpening when multiple policies induce similar observations**. The abstracted latent types in QOM allow for faster and more stable belief evolution within the search tree, which can lead to better planning decisions.
> >
> > > In Figure 4(b), Best-k performance is very similar to QOM. Given the amount of overlap between the curves, it is not possible to draw any conclusions from this plot.
> >
> > We would like to clarify that the Best-k and Worst-k baselines are not realizable strategies, but rather hypothetical upper and lower bounds. Specifically, Best-k reports the highest return achieved by selecting the optimal single opponent type $k$ in hindsight, while Worst-k reports the lowest. These baselines are constructed by evaluating all possible latent types independently and selecting the best (or worst) performing result post hoc.
> >
> > The intention of including Best-k in the plot is to demonstrate that even the best fixed choice of type cannot consistently outperform the belief-weighted marginalization employed by QOM. This highlights a key insight of our method: selecting a single opponent type—even the best one in hindsight—is insufficient under uncertainty, and instead, structured belief-aware reasoning is essential for robust performance.
> >
> > To avoid potential confusion, we will revise it to make this distinction more explicit. We will also clarify that Best-k and Worst-k serve only as performance envelopes to contextualize the value of QOM’s principled Bayesian method.
> >
> > > It is unclear why sampling in every state is necessary to achieve this. If multiple rollouts are run, sampling at the root node still marginalizes over different opponent types.
> >
> > Clarification of the previous question also helps address this question. While sampling only at the root may produce an approximate marginalization over opponent types across rollouts, it does not account for the evolving posterior belief as new (simulated) opponent actions are observed during each trajectory. QOM performs dynamic belief updates at every step, allowing the agent to refine its understanding of the opponent and adapt its policy accordingly within the same planning simulation. This recursive belief refinement is a core advantage of our approach and contributes significantly to planning effectiveness.
> >
> > To validate this, we conducted an additional ablation study comparing QOM with dynamic belief updates at each simulation step (QOM-step, our default method) versus a version that samples an opponent type only once at the root and uses it throughout the rollout (QOM-root). As shown in the table below:
> >
> > |Search Time|0.1|1|5|10|20|
> > |---|---|---|---|---|---|
> > |QOM-step|-3.18 ± 1.40|-2.94 ± 1.50|-1.46 ± 1.58|0.99 ± 1.96|1.59 ± 1.58|
> > |QOM-root|-4.35 ± 2.02|-4.18 ± 2.00|-3.52 ± 1.51|-2.99 ± 1.23|-2.74 ± 1.21|
> >
> > The results show a clear performance gap. Crucially, this gap cannot be explained merely by belief updates—it arises because QOM-step treats the opponent not as a fixed type but as a mixture over latent types throughout the rollout. In contrast, QOM-root effectively commits to a single hypothetical opponent type, missing the opportunity to account for uncertainty during simulation.

---

> > > ### Author Response · Authors · 2025-08-03
> > > **Response to Reviewer ETPj （Part Ⅱ）**
> > >
> > > >More importantly, none of these reasons directly relate to the central idea of quantized opponent models. Including the meta-policy confounds the analysis, making it difficult to assess how much of the performance gain is due to the quantized models themselves versus the meta-policy.
> > >
> > > We appreciate the reviewer’s concern about whether the meta-policy component is separable from the core contribution of quantized opponent models. From the perspective of quantized opponent policies, our comparison with baselines such as POTMMCP and the reviewer-suggested QOM-direct provides clear insight. We would like to thank the reviewer again for suggesting QOM-direct—it has helped us improve the clarity and completeness of the work.
> > >
> > > It is also worth noting that the idea of using a meta-policy was introduced by POTMMCP. Our method builds upon this concept and integrates it with a learned quantized opponent model. Therefore, when comparing QOM to POTMMCP, both methods leverage belief-aware meta-policies, but the performance gap stems from QOM’s quantization-based opponent abstraction. This highlights that the main contribution of QOM lies not in the meta-policy design per se, but in enabling scalable and tractable belief-aware planning through quantized opponent models.
> > >
> > > To further isolate the role of the meta-policy, we include an ablation study in which QOM is used without the meta-policy, using a uniform rollout strategy. As shown in the table below, removing the meta-policy leads to degraded performance across all search budgets:
> > >
> > > |Search Time|0.1|1|5|10|20|
> > > |---|---|---|---|---|---|
> > > |QOM (with meta-policy)|-3.18 ± 1.40|-2.94 ± 1.50|-1.46 ± 1.58|0.99 ± 1.96|1.59 ± 1.58|
> > > |QOM (w/o meta-policy)|-4.53 ± 1.25|-3.72 ± 1.57|-2.02 ±  1.89|0.24 ± 1.72|1.42 ± 1.45|
> > >
> > > This result shows that while the meta-policy is important for realizing the benefits of QOM in practice, the underlying source of improvement comes from the quantized representation of opponent policies. The meta-policy enables belief-aware planning, but it is the opponent quantization that makes this planning both scalable and effective.
> > >
> > > We hope that the supplementary experiments and explanations provided have addressed your concern.

---

> > ### Author Response · Authors · 2025-08-06
> > **To reviewer ETPj**
> >
> > Has my response addressed your concerns? If there are any remaining issues, please let me know. If everything is clear, could you consider adjusting the score? Thank you sincerely for your review.

---

> > > ### Comment · Reviewer_ETPj · 2025-08-06
> > >
> > > I would like to thank the authors for clarifying why sampling opponents at every state is a good idea. This now makes sense to me.
> > >
> > > I appreciate the new results showing the performance of QOM with and without the meta-policy. However, additional experiments are needed to better evaluate the benefits of QOM in isolation from the meta-policy. Given this, along with my concerns about the weak experimental results (e.g., overlapping error bars), I cannot recommend the paper for acceptance.

---

> > > > ### Author Response · Authors · 2025-08-06
> > > > **Response to Reviewer ETPj （Round 2 Part Ⅰ）**
> > > >
> > > > We thank the reviewer for acknowledging the rationale behind per-state opponent sampling, and for the constructive suggestions regarding meta-policy isolation and error bars. As follows, we address your concerns in detail.
> > > >
> > > > > Given this, along with my concerns about the weak experimental results (e.g., overlapping error bars).
> > > >
> > > > ## Response 1: On overlapping error bars
> > > >
> > > > * **Error-bar overlap does not imply weak evidence—and it appears only in a small subset of settings.** We acknowledge that a few plots display overlapping error bars; however, these cases constitute a minor portion of our full experimental suite, and several of them are ablation figures whose purpose is sensitivity analysis rather than asserting large performance gaps. Concretely, we evaluate 8 settings (Fig. 2a–f, Fig. 3a–b) × 3 competitive baselines (POTMMCP, I-POMCP-PF, MBOM) × 5 search budgets (0.1, 1, 5, 10, 20), i.e., 120 method–budget comparisons; each algorithm–setting–budget cell is averaged over 300 independent runs (and 1,000 runs for the extended RWS tables) which demonstrates that our empirical study is substantial and robust, not dependent on a few isolated points. Across these 120 comparisons, only 8 show overlapping error bars. Overall, QOM is superior or non-inferior to the baselines in the vast majority of settings. By contrast, Fig. 3(c) and Fig. 4(a–d) are ablations and are not used to substantiate SOTA claims; any overlap there reflects sensitivity analysis rather than the main result. Taken together, these facts do not support the assertion that *"the weak experimental results (e.g., overlapping error bars)"*
> > > >
> > > > * **We increased the number of runs to tighten the intervals.** To directly address the error-bar overlap concern, we raised the sampling to 1,000 independent runs per condition, which materially narrows the intervals and stabilizes the estimates. We previously provided the 1,000-run extension for Fig. 3(a) (Table 1 in our earlier response); here we add the new 1,000-run extension for Fig. 3(b) (Table 2). Across both tables, the intervals are tighter while the advantages are preserved. In the revision, we will include the full set of extended tables.
> > > >
> > > > Table: Extended results for Figure 3(a) — Adaptation evaluation in RWS with 1000 runs
> > > > |Search Time|0.1|1|5|10|20|
> > > > |---|---|---|---|---|---|
> > > > |QOM|-3.91 ± 0.94|-2.82 ± 1.09|-1.53 ± 0.80|-0.09 ± 0.80|0.20 ± 1.07|
> > > > |POTMMCP|-5.11 ± 1.08|-4.32 ± 0.79|-2.66 ± 0.89|-1.44 ± 1.01|-1.51 ± 0.90|
> > > > |I-POMCP-PF|-5.60 ± 0.80|-4.34 ± 1.15|-3.85 ± 0.93|-3.54 ± 0.95|-2.93 ± 0.93|
> > > > |MBOM|-4.08 ± 0.79|-3.15 ± 1.07|-1.44 ± 0.92|-1.03 ± 1.01|0.19 ± 0.95|
> > > >
> > > > Table2: Extended results for Figure 3(b) — Generalization evaluation in RWS with 1000 runs
> > > > |Search Time|0.1|1|5|10|20|
> > > > |---|---|---|---|---|---|
> > > > |QOM|-4.19 ± 0.94|-3.48 ± 1.06|-2.25 ± 0.83|0.05 ± 1.01|0.92 ± 0.96|
> > > > |POTMMCP|-4.1 ± 1.19|-3.7 ± 0.93|-3.3 ± 0.85|-2.27 ± 1.04|0.02 ± 0.82|
> > > > |I-POMCP-PF|-6.31 ± 0.87|-5.91 ± 0.82|-5.37 ± 1.02|-4.0 ± 0.82|-2.42 ± 0.84|
> > > > |MBOM|-4.59 ± 0.82|-3.63 ± 1.07|-2.26 ± 0.95|-0.8 ± 0.93|0.26 ± 1.05|
> > > >
> > > > We respectfully disagree that the results are "weak." The overlap occurs in only 8 of 120 method–budget comparisons. Our evaluation is large-scale and robust (8 settings × 3 baselines × 5 budgets, with 300 runs per configuration), and the extended tables show tighter confidence intervals while preserving the observed advantages. These facts do not support the claim of "weak experimental results (e.g., overlapping error bars)."

---

> ### Author Response · Authors · 2025-08-06
> **Response to Reviewer ETPj （Round 2 Part Ⅱ）**
>
> > However, additional experiments are needed to better evaluate the benefits of QOM in isolation from the meta-policy.
>
> ## Response 2: On the role of the meta-policy
>
> * **Meta-policy should not be removed from the main evaluation.**
>
>   * **It is integral to the QOM framework by design.** QOM is a unified modeling–inference–planning framework with three tightly coupled components: (i) quantized opponent representation $\rightarrow$ (ii) step-wise posterior updates inside rollouts $\rightarrow$ (iii) a meta-policy that maps the belief to a PUCT prior and rollout guidance. It is important that meta-policy is uniquely tailored to QOM: it is constructed as a belief-weighted mixture of soft best responses to the learned latent opponent types and then used for planning, not a generic prior. Section 3.3 further defines $\pi_i^{\text{meta}}$ as $\pi^{\text{meta}}\_i(a | h)=\sum_{k} b\_t(k)\sum_{l}\sigma(\pi_i^{(l)} | k)\,\pi_i^{(l)}(a | h)$, where $\sigma(\pi_i^{(l)} | k)$ is a soft best-response based on a latent-type payoff matrix $R_{l,k}$ estimated from trajectories labeled by the quantized autoencoder—this ties the meta-policy's form directly to QOM's latent-type abstraction. The meta-policy is the planning interface of QOM—removing it changes the algorithm and evaluates something other than QOM. Our claims concern the complete QOM framework as deployed under realistic online-planning budgets.
>
>   * **Methodological fairness and consistency with prior art.** Leading opponent-aware planners (e.g., POTMMCP) also use belief-aware priors/rollouts. Removing QOM's meta-policy while baselines retain theirs would be asymmetric.
>
>   * For example, removing the meta-policy is like evaluating AlphaZero without its policy/value networks—an instructive ablation, but not the system being claimed. The scientifically relevant question is the performance of the complete algorithm under compute constraints.
>
>
> * **Component-isolation evidence addressing the request.**
>
>   While we argue it should not be stripped from the main evaluation, we nonetheless provide ablations to isolate the meta-policy's role:
>
>   * **QOM without the meta-policy (uniform rollouts).** We add an ablation study that removes the meta-policy and uses uniform rollouts while keeping the quantized representation and step-wise posterior updates unchanged, to isolate dependence on meta-policy.
>
>   * **Same-meta-policy control: QOM vs. QOM-direct.** We add an ablation study in which both methods use the same meta-policy; we compare QOM (quantized $K$ types) with QOM-direct (exact Bayesian updates over all 50 policies), to isolate the contribution of the opponent representation and inference dynamics apart from the planning prior. We will include the corresponding results in the revision.
>
> Table: Meta-Policy Ablation in RWS
> |Search Time|0.1|1|5|10|20|
> |---|---|---|---|---|---|
> |QOM |-3.18 ± 1.40|-2.94 ± 1.50|-1.46 ± 1.58|0.99 ± 1.96|1.59 ± 1.58|
> |QOM (w/o meta-policy)|-4.53 ± 1.25|-3.72 ± 1.57|-2.02 ± 1.89|0.24 ± 1.72|1.42 ± 1.45|
>
> Table: Performance comparison between QOM and QOM-direct in PE(Evader)
> |Search Time|0.1|1|5|10|20|
> |---|---|---|---|---|---|
> |QOM|0.10 ± 0.08|0.59 ± 0.07|0.69 ± 0.08|0.69 ± 0.07|0.72 ± 0.10|
> |QOM-direct|-0.26 ± 0.09|0.01 ± 0.06|0.06 ± 0.10|0.12 ± 0.09|0.16 ± 0.09|
>
> The meta-policy is integral to QOM and removing it would evaluate a different algorithm and create an asymmetric comparison to baselines. To address the request, we include two ablation studies: (i) QOM without the meta-policy (uniform rollouts), and (ii) a same-meta-policy control comparing QOM to QOM-direct. Taken together, we believe the paper already evaluates QOM both as a complete framework and under component isolation; accordingly, further meta-policy–only isolation beyond what we report does not appear essential for assessing the contribution.

---

### Note · Authors · 2025-08-12

We thank the AC and reviewers for their time, thoughtful questions, and constructive feedback. We sincerely appreciate the recognition of QOM's technical strength and core contributions. Below we summarize how we have addressed the main concerns:

* Empirical robustness & evaluation clarity. For Reviewer ```ETPj```, the error-bar overlap issue was handled by raising repetitions to 1,000 (tighter CIs) and tempering claims. In response to comments from ```VDJ5``` and ```PHgn```, the "search time" plot will be redrawn using a categorical axis; it is clarified that all methods were run under identical wall-clock budgets.

* Mechanism vs. tricks: new baseline and ablations. Per Reviewer ```ETPj```, we added an exact-Bayes comparator (QOM-direct). Results show QOM clearly stronger at low budgets and convergent at high budgets. Ablations demonstrate that removing the meta-policy consistently degrades performance.

* Contribution and novelty. In response to Reviewer ```PHgn```, we position QOM as requiring no explicit goal space and applicable to general partially observable multi-agent settings. The Related Work will be expanded with a contrast to HOP.

* Efficiency claim—scope and boundaries. Addressing Reviewer ```PHgn```'s concern, we make explicit that offline learning costs are out of scope; our claim is higher effectiveness under a fixed search-time budget, not free speedups. These boundaries will be stated clearly.

* Reproducibility and setup clarity. For Reviewer ```PqEB``` and ```PHgn```, we specify which domains use real vs. learned simulators (aligned with prior work) and provide architecture and training hyperparameters. In line with reproducibility requests, we will release code in the revision.

* Type–policy correspondence. Responding to Reviewers ```VDJ5``` and ```PqEB```, we add visualizations and correspondence checks and clarify that quantization induces a structured opponent space, reducing reliance on hand-crafted classes; these materials will be included in the revision.

**On error-bar overlap and the meta-policy, we provided detailed analyses in "Response to Reviewer ```ETPj``` (Round 2)." Although there was no follow-up, we respectfully ask the AC to consider that exchange.**

Our thanks once more to the AC and reviewers for their careful evaluation and valuable comments. We are confident that every concern, including those prompting further experiments, is resolved in the rebuttal and will be integrated into the revision.

---

### Decision · Program_Chairs · 2025-09-17

**Decision:**

Accept (poster)

**Comment:**

This uses a hybrid approach, where a finite number of opponents are model with neural networks, and a Bayesian prior is maintained over them.

+ The approach is interesting
- It is a bit complicated. Maybe a simpler version of the algorithm should have been tested as well (which the authors do in their rebuttal)
- The experimental results are not very convincing, i.e. the algorithm is not strictly better than the baseline.

The reveiwers are mainly positive. ETPj asks for some more experiments, but it is unclear to me what those would be.